 **RESEARCH ARTICLE**

# Miro1-dependent mitochondrial dynamics in parvalbumin interneurons

**Georgina Kontou[1], Pantelis Antonoudiou[2], Marina Podpolny[3], Blanka R Szulc[1], I Lorena Arancibia-Carcamo[1], Nathalie F Higgs[1], Guillermo Lopez-Domenech[1], Patricia C Salinas[3], Edward O Mann[2,4], Josef T Kittler[1]***

[1]Department of Neuroscience, Physiology and Pharmacology, University College London, London, United Kingdom; [2]Department of Physiology, Anatomy and Genetics, University of Oxford, Oxford, United Kingdom; [3]Department of Cell and Developmental Biology, University College London, London, United Kingdom; [4]Oxford Ion Channel Initiative, University of Oxford, Oxford, United Kingdom

**Abstract** The spatiotemporal distribution of mitochondria is crucial for precise ATP provision and calcium buffering required to support neuronal signaling. Fast-spiking GABAergic interneurons expressing parvalbumin (PV+) have a high mitochondrial content reflecting their large energy utilization. The importance for correct trafficking and precise mitochondrial positioning remains poorly elucidated in inhibitory neurons. Miro1 is a $Ca^{2+}$-sensing adaptor protein that links mitochondria to the trafficking apparatus, for their microtubule-dependent transport along axons and dendrites, in order to meet the metabolic and $Ca^{2+}$-buffering requirements of the cell. Here, we explore the role of Miro1 in PV+ interneurons and how changes in mitochondrial trafficking could alter network activity in the mouse brain. By employing live and fixed imaging, we found that the impairments in Miro1-directed trafficking in PV+ interneurons altered their mitochondrial distribution and axonal arborization, while PV+ interneuron-mediated inhibition remained intact. These changes were accompanied by an increase in the ex vivo hippocampal γ-oscillation (30–80 Hz) frequency and promoted anxiolysis. Our findings show that precise regulation of mitochondrial dynamics in PV+ interneurons is crucial for proper neuronal signaling and network synchronization.

**\*For correspondence:**
j.kittler@ucl.ac.uk

**Competing interests:** The authors declare that no competing interests exist.

## Introduction

Parvalbumin (PV+) interneurons constitute a small proportion of the total neuronal population (less than 2% in the hippocampus), yet they possess crucial roles in shaping neuronal network activity (*Freund and Buzsáki, 1996*; *Jonas et al., 2004*; *Pelkey et al., 2017*). PV+ interneurons inhibit their postsynaptic targets efficiently by applying fast perisomatic inhibition and have been directly implicated in the generation of network activity at the gamma (γ) band frequency (30–80 Hz) (*Antonoudiou et al., 2020*; *Cardin et al., 2009*; *Hájos et al., 2004*; *Mann et al., 2005*; *Sohal et al., 2009*). Network oscillations at γ-band frequency are believed to facilitate information transmission through circuit synchronization and local gain control that may be instrumental in multiple cognitive processes such as attention, learning, and memory (*Akam and Kullmann, 2010*; *Fries, 2015*; *Howard et al., 2003*; *Montgomery and Buzsaki, 2007*; *Sohal, 2016*). Importantly, these oscillations are thought to be metabolically very costly, and it has therefore been postulated that PV+ interneurons require substantial amounts of energy via ATP hydrolysis to sustain the high firing rate and dissipate ion gradients during neuronal transmission (*Attwell and Laughlin, 2001*; *Kann, 2011*; *Kann, 2016*; *Kann and Kovács, 2007*; *Kann et al., 2014*). Thus, it is crucial to understand the metabolic expenditure and the involvement of mitochondria in PV+ interneurons. Indeed, electron microscopy, histochemical, and transcriptomic approaches have revealed that PV+ interneurons have a higher density of energy-producing mitochondria and elevated expression levels of electron

transport chain components (*Adams et al., 2015*; *Gulyás et al., 2006*; *Nie and Wong-Riley, 1995*; *Paul et al., 2017*).

The spatiotemporal organization of mitochondria is essential for the precise provision of ATP and $Ca^{2+}$-buffering for neuronal transmission and communication (*Devine and Kittler, 2018*; *MacAskill and Kittler, 2010*). Miro1 is a mitochondrial adaptor protein, responsible for coupling mitochondria to the cytoskeleton and for their bidirectional trafficking in axons and dendrites (*Birsa et al., 2013*; *Guo et al., 2005*; *López-Doménech et al., 2016*; *López-Doménech et al., 2018*; *Macaskill et al., 2009*; *Nguyen et al., 2014*; *Saotome et al., 2008*; *Wang and Schwarz, 2009*). Global deletion of Miro1 (encoded by the *Rhot1* gene) is perinatal lethal, while the conditional removal of Miro1 from cortical and hippocampal pyramidal cells alters the occupancy of dendritic mitochondria due to impairment in trafficking, resulting in dendritic degeneration and cell death (*López-Doménech et al., 2016*). In contrast, the significance of mitochondrial trafficking and distribution in PV+ interneurons, and the role of Miro1, is completely unexplored and especially interesting as their axon is highly branched with a cumulative length reaching up to 50 mm in the hippocampus (*Hu et al., 2014*).

In this study, we generated a transgenic mouse line where mitochondria are fluorescently labelled in PV+ interneurons. We crossed this line with the Miro1 floxed mouse (*Rhot1*^flox/flox^), to generate a model where Miro1 was conditionally knocked-out exclusively in PV+ interneurons. Using two-photon live-imaging of ex vivo organotypic brain slices, we demonstrated a reduction in mitochondrial trafficking in the absence of Miro1 in PV+ interneurons in the hippocampus. The impairment in Miro1-directed mitochondrial transport led to an accumulation of mitochondria in the soma and their depletion from axonal presynaptic terminals in acute hippocampal brain slices. Loss of Miro1 resulted in alterations in axonal but not dendritic branching in PV+ interneurons. While the ability of PV+ interneurons to apply long-lasting inhibition to post-synaptic targets remained intact, the changes in Miro1-dependent mitochondrial dynamics were accompanied by an increased frequency of γ-oscillations in hippocampal brain slices and a reduction in anxiety-related emotional behavior. Thus, we show that Miro1-dependent mitochondrial positioning is essential for correct PV+ interneuron function, network activity, and anxiolytic animal behavior.

## Results

### Loss of Miro1 in parvalbumin interneurons impairs mitochondrial trafficking

Mitochondrial enrichment in parvalbumin (PV+) interneurons is thought to reflect the high energetic demands of these cells (*Gulyás et al., 2006*). Consistent with this finding, we also observed that the protein levels of subunit IV of cytochrome c oxidase (COX-IV) were elevated in PV immuno-positive regions ($3.6 \times 10^{-6} \pm 1.33 \times 10^{-5}$ a.u.) when compared to immuno-negative areas in hippocampal slices ($2.8 \times 10^{-6} \pm 1.26 \times 10^{-5}$ a.u., *Figure 1A*, p<0.0001, Mann–Whitney U test), further supporting that PV+ interneurons rely heavily on mitochondria. Yet, mitochondrial dynamics in PV+ interneurons have not been explored. To specifically examine the role of mitochondrial transport in PV+ interneurons, we disrupted the mitochondrial adaptor protein Miro1 by crossing the *Pvalb*^Cre^ mouse line (*Hippenmeyer et al., 2005*) with the *Rhot1*^flox/flox^ mouse (*López-Doménech et al., 2016*), thus generating a model where Miro1 was selectively knocked-out in PV+ interneurons (*Figure 1—figure supplement 1A*). We then crossed this *Pvalb*^Cre^ *Rhot1*^flox/flox^ line with a transgenic mouse where mitochondria expressed the genetically encoded Dendra2 fluorophore (MitoDendra) (*Pham et al., 2012*; *Figure 1—figure supplement 1A*), allowing for the visualization of mitochondria selectively in PV+ interneurons (*Figure 1—figure supplement 1B*). Thus, we generated a mouse where Miro1 was knocked-out and mitochondria were fluorescently labelled selectively in PV+ interneurons (*Figure 1—figure supplement 1A,D*). Gene expression under the PV+ promoter begins around postnatal day 10 (P10) and is stabilized around P28 (*Barnes et al., 2015*). Thus, Cre-dependent removal of Miro1 is not expected to impact on the migration, differentiation, and viability of PV+ neurons (*Okaty et al., 2009*; *del Río et al., 1994*). Indeed, the number of MitoDendra-expressing PV+ cells did not vary in the hippocampus of control (Miro1 WT: *Pvalb*^Cre^ *Rhot1*^+/+^), hemi-floxed (Miro1 HET: *Pvalb*^Cre^ *Rhot1*^flox/+^), and conditional knock-out (Miro1 KO: *Pvalb*^Cre^ *Rhot1*^flox/flox^) animals (*Figure 1—figure supplement 1C*). The Miro1 fluorescence intensity was significantly reduced in PV+

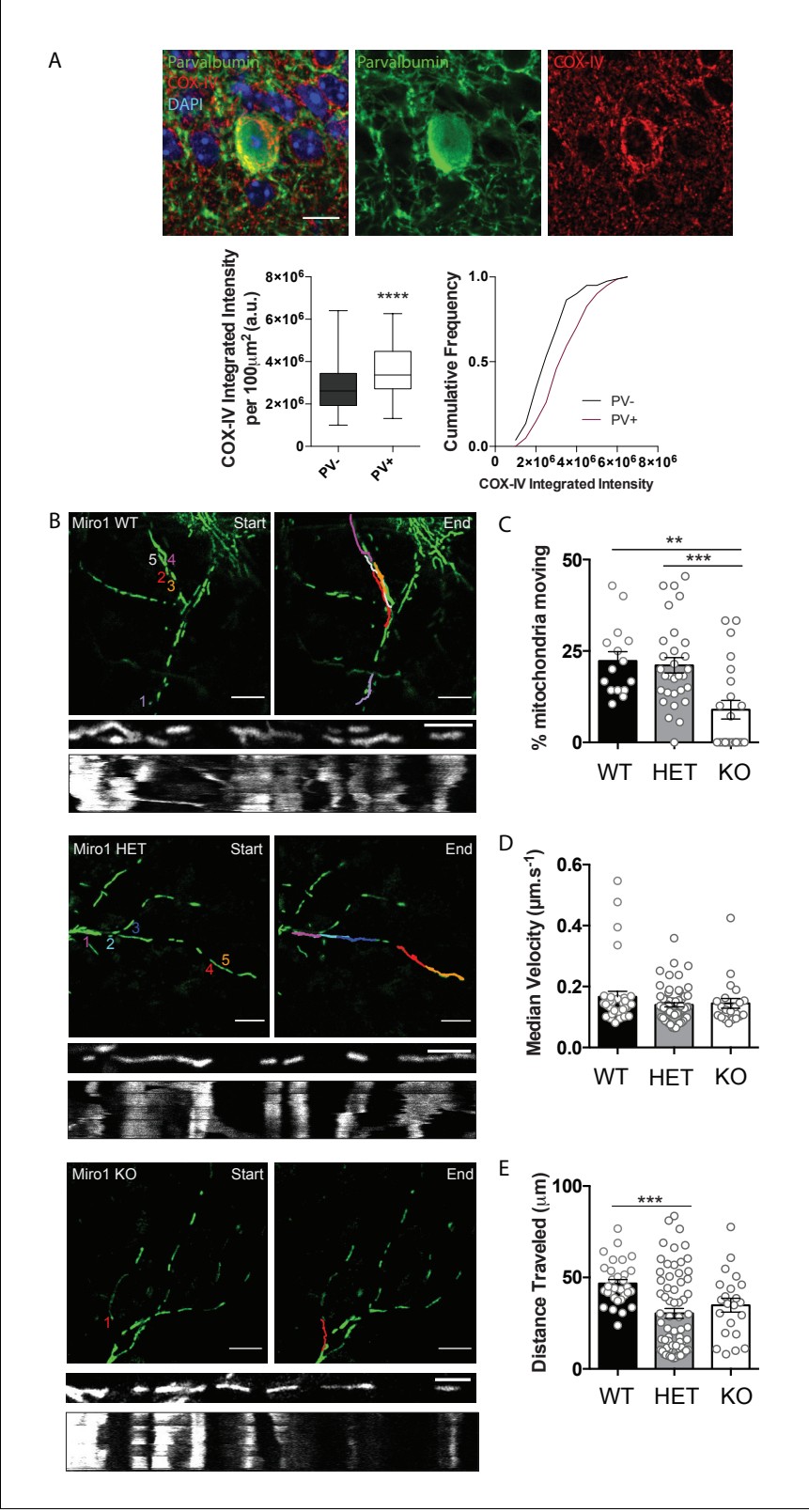

**Figure 1.** Cell-type specific removal of Miro1 from parvalbumin interneurons impairs mitochondrial trafficking.
(**A**) COX-IV levels in PV+ interneurons. The fluorescence signal of COX-IV is increased in PV immuno-positive cells in the hippocampus. Scale bar = 10 μm. Boxplot and cumulative distribution show the quantification of the mean integrated intensity of the COX-IV fluorescent signal in PV immuno-positive and -negative regions (n = 81 cells, 11

*Figure 1 continued on next page*

*Figure 1 continued*

slices, three animals). (**B**) Miro1-directed mitochondrial trafficking in organotypic brain slices. Representative images from a 500 s two-photon movie of MitoDendra+ mitochondria in PV+ interneurons in ex vivo organotypic hippocampal slices from WT, HET, and Miro1 KO animals. The colored numbers/lines denote tracks of individual mobile mitochondria during the movie. Scale bar = 10 μm. The kymographs represent the mitochondrial displacement over time. Scale bar = 5 μm. (**C**) Quantification of the percentage of moving mitochondria ($n_{WT}$ = 15 movies, nine slices, four animals, $n_{HET}$ = 32 movies, 10 slices, four animals, $n_{KO}$ = 22 movies, eight slices, four animals). (**D**) Quantification of the median velocity ($n_{WT}$ = 35 mitochondria, $n_{HET}$ = 65 mitochondria, $n_{KO}$ = 22 mitochondria). (**E**) Quantification of the distance traveled ($n_{WT}$ = 31 mitochondria, $n_{HET}$ = 66 mitochondria, $n_{KO}$ = 22 mitochondria).

The online version of this article includes the following source data and figure supplement(s) for figure 1:

**Source data 1.** Source data for COX-IV levels and mitochondrial trafficking in PV+ interneurons.
**Figure supplement 1.** Generation of the *Pvalb$^{Cre}$ Rhot1$^{flox/flox}$* MitoDendra transgenic mouse line and conditional removal of Miro1 from MitoDendra-expressing parvalbumin interneurons.
**Figure supplement 1—source data 1.** Source data for cell viability and Miro1 levels.

interneurons from brain slices of knock-out mice, confirming the selective removal of Miro1 from PV + interneurons (HET 844 ± 32.8 a.u., KO 518 ± 20.5 a.u. *Figure 1—figure supplement 1D*, p<0.0001, Mann–Whitney U test). These immunohistochemical experiments demonstrate that the *Pvalb$^{Cre}$ Rhot1$^{flox/flox}$* MitoDendra mouse can be utilized to examine mitochondrial dynamics in PV+ interneurons in a system where Miro1 is absent.

Next, we wanted to investigate the contribution of Miro1 to mitochondrial trafficking in PV+ interneurons. We therefore performed two-photon live-imaging in neuronal processes of intact organotypic brain tissue (at 7–9 days in vitro) from neonatal (P6–8) Miro1 WT, HET, and KO animals (*Figure 1B*). Consistent with other models where Miro1 is knocked-out (*López-Doménech et al., 2016*; *Nguyen et al., 2014*), we found a significant reduction in the percentage of moving mitochondria in the absence of Miro1 when compared to the control (WT 23 ± 3%, HET 21 ± 2%, KO 9 ± 3%, *Figure 1C*, $p_{HET}$ = 0.939, $p_{KO}$ = 0.002, Tukey's multiple comparisons test, ordinary one-way ANOVA F(2, 64) = 9.107, P = 0.0003). Even though the moving mitochondria exhibited similar velocities in the three conditions (WT 0.17 ± 0.018 μm/s, HET 0.14 ± 0.007 μm/s, KO 0.14 ± 0.016 μm/s, *Figure 1D*, $p_{HET}$ = 0.229, $p_{KO}$ = 0.557, Tukey's multiple comparisons test, ordinary one-way ANOVA F(2, 119) = 1.389, P = 0.2534), the length of the traveled trajectory was shorter in HET (31 ± 2.7 μm) and KO (35 ± 3.8 μm) when compared to the control (47 ± 2.1 μm, *Figure 1E*, $p_{HET}$ = 0.0004, $p_{KO}$ = 0.070, Tukey's multiple comparisons test, ordinary one-way ANOVA F(2, 116) = 7.727, P = 0.0007). This experiment demonstrates that the presence of Miro1 is critical for mitochondrial trafficking in PV+ interneurons.

## Loss of Miro1 in parvalbumin interneurons results in mitochondrial clustering in the soma and depletion from axonal presynaptic terminals

Given that Miro1 is a crucial component of the mitochondrial transport machinery in PV+ cells, we wanted to investigate the effect of impaired trafficking on mitochondrial localization in PV+ interneurons, ex vivo. We noticed that mitochondria clustered in the cell bodies of PV+ interneurons depleted of Miro1 in hippocampal sections from adult mice (*Figure 2A*). The number of PV immunopositive cells that contained mitochondrial clusters in the soma was significantly increased when Miro1 was knocked-out (58 ± 5.7%), compared to the hemi-floxed condition (11 ± 3.1%, *Figure 2B*, p < 0.0001, unpaired t-test). This is consistent with the somatic mitochondrial clusters that have also been reported in pyramidal cells in the *Camk2a$^{Cre}$ Rhot1$^{flox/flox}$* model (*López-Doménech et al., 2016*) and are presumably due to the inefficient trafficking of mitochondria out to the neurites of the cell. The mitochondria in the Miro1 KO cells occupied a smaller area relative to the cell body (HET 39 ± 1.0%, KO 28 ± 1.0%) and seem to be concentrated around the nucleus when Miro1 was knocked-out (*Figure 2C*, p<0.0001, unpaired t-test). Together, these data suggest that the spatial distribution of mitochondria is altered in PV+ interneurons in the absence of Miro1.

To further understand the impact of impaired Miro1-dependent mitochondrial trafficking in PV+ interneurons, we looked at the organization of mitochondria in the axons and dendrites of PV+ cells. MitoDendra+ cells were biocytin filled allowing for the morphological visualization of individual PV+

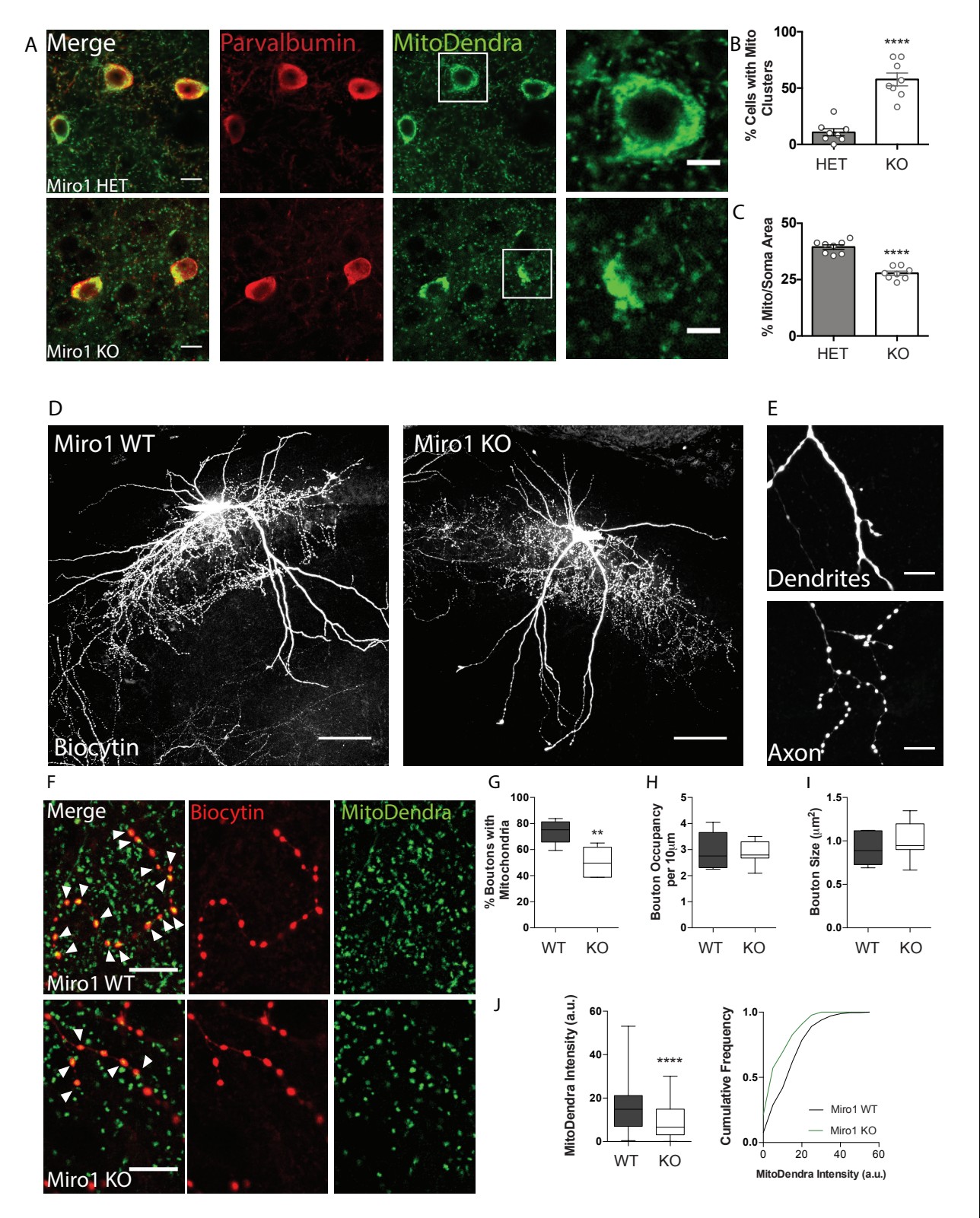

**Figure 2.** Loss of Miro1 results in an accumulation of mitochondria in the soma and depletion from axonal presynaptic terminals. (**A**) Loss of Miro1 results in MitoDendra+ clustering in the somata of PV+ interneurons. Confocal images from HET and Miro1 KO PV immuno-positive cells in fixed brain tissue. Scale bar = 10 µm. Cells in the white box are zoomed as illustrated in the images on the right, to clearly depict the mitochondrial clusters in the Miro1 KO cells. Scale bar = 5 µm. (**B**) Bar chart shows the quantification for the percentage of cells that contain mitochondrial clusters ($n_{HET}$ = eight

*Figure 2 continued on next page*

*Figure 2 continued*

slices, three animals and $n_{KO}$ = eight slices, three animals). (C) Quantification for the percentage area that mitochondria occupy within the PV immuno-positive soma. Due to the presence of mitochondrial clusters the %Mito/Soma area is significantly decreased in the Miro1 KO neurons ($n_{HET}$ = eight slices, three animals and $n_{KO}$ = eight slices, three animals). (D) Representative max-projected confocal stack of biocytin-filled PV+ interneurons in the hippocampus of 350 μm fixed acute brain slices. Scale bar = 100 μm. (E) Example of the distinct distribution of biocytin in dendritic and axonal compartments. Biocytin is diffused in dendrites (top) and clustered in axonal terminals (bottom). (F) Loss of Miro1 depletes mitochondria from axonal presynaptic terminals. Representative images of biocytin-filled synaptic boutons (red) and mitochondria (green) in WT and Miro1 KO neurons. Arrows point to boutons that contain mitochondria. Scale bar = 5 μm. (G) Boxplots for the quantification of the percentage of boutons that contain mitochondria ($n_{WT}$ = five neurons, four slices, two animals and $n_{KO}$ = seven neurons, six slices, three animals). (H) Boxplots for the quantification of the occupancy of boutons in axonal segments ($n_{WT}$ = five neurons, four slices, two animals and $n_{KO}$ = seven neurons, six slices, three animals). (I) Quantification of the mean size of boutons ($n_{WT}$ = 625 boutons, five neurons, four slices, two animals, $n_{KO}$ = 526 boutons, seven neurons, six slices, three animals). (J) Boxplot and cumulative distribution for the quantification of the mean MitoDendra fluorescent intensity within biocytin-filled puncta ($n_{WT}$ = 246 boutons, $n_{KO}$ = 173 boutons).

The online version of this article includes the following source data for figure 2:

**Source data 1.** Source data for MitoDendra+ mitochondria in the soma and axonal presynaptic terminals of PV+ interneurons.

interneurons in acute brain slices (*Figure 2D*). Biocytin was uniformly diffused along dendrites and accumulated in the axon allowing for the identification of distinct presynaptic boutons (*Figure 2E, F*; *Swietek et al., 2016*). We observed that 74 ± 4.1% of presynaptic terminals contained MitoDendra+ mitochondria in control cells (*Figure 2F,G*). This is consistent with literature where approximately 75% of PV+ axonal boutons are enriched with mitochondria, and this is in contrast to pyramidal cells where less than half are associated with mitochondria (*Glausier et al., 2017*; *Kwon et al., 2016*; *Smith et al., 2016*; *Vaccaro et al., 2017*). However, we report a marked reduction in the axonal presynaptic terminals that contained mitochondria in the cells where Miro1 was knocked-out (51 ± 4.0%, *Figure 2G*, p = 0.003, unpaired t-test). In agreement with this, the MitoDendra fluorescence intensity within the biocytin-filled puncta was reduced in the Miro1 KO compared to WT (WT 15.2 ± 0.64 a.u., KO 9.2 ± 0.60 a.u. *Figure 2J*, p < 0.0001, Mann–Whitney U test). The distribution along the axon (occupancy per 10 μm) and size of boutons were unaffected by the loss of Miro1 (*Figure 2H,I*). These data indicate that defects in Miro1-directed mitochondrial trafficking are sufficient to shift the mitochondrial distribution along the axon, leading to a depletion of mitochondria away from axonal presynaptic boutons.

## Loss of Miro1 in PV+ interneurons results in an enrichment of mitochondria close to axonal branching sites and an increase in axonal branching

Our data suggest that the impairment in Miro1-dependent trafficking altered the precise mitochondrial positioning in PV+ interneuron axons. To examine whether this alteration could result in a structural change in PV+ interneuron morphology due to a reorganization of the mitochondrial network in the absence of Miro1, we used the biocytin fill to generate a neuronal reconstruction (*Figure 3A*). The total length of PV+ interneuron dendrite and axon did not differ between control and conditional knock-out cells (*Figure 3B*). We noticed an increase in the number of processes in Miro1 KO cells (506 ± 34 processes) when compared to WT (367 ± 46 processes, *Figure 3C*, p=0.042, unpaired t-test) and a trend towards an increase in the number of branches between WT (342 ± 42 branch points) and Miro1 KO conditions (465 ± 35 branch points, *Figure 3D*, p=0.052, unpaired t-test). We further investigated whether the increase in processes was attributed to a selective increase in the branches of axons or dendrites in the absence of Miro1. By visually assessing and independently labelling axons and dendrites during the reconstruction process, we isolated the contribution of each compartment to the total number of branches. Interestingly, there was a selective increase in axonal (*Figure 3F,H*), but not dendritic branching (*Figure 3E,G*) in Miro1 KO (427 ± 32 branch points) when compared to WT (296 ± 38 branch points, *Figure 3F*, p=0.031, unpaired t-test). By performing Sholl analysis and plotting the number of intersections as a function of the distance from the soma in WT and Miro1 KO cells, we noticed an enhancement in axonal branching in the proximal axon (between 50 μm and 200 μm from the cell body) when Miro1 was knocked-out (*Figure 3H*). By using the neuronal reconstruction as a mask, we isolated the MitoDendra+ mitochondrial network in individual PV+ interneurons and compared the mitochondrial organization in

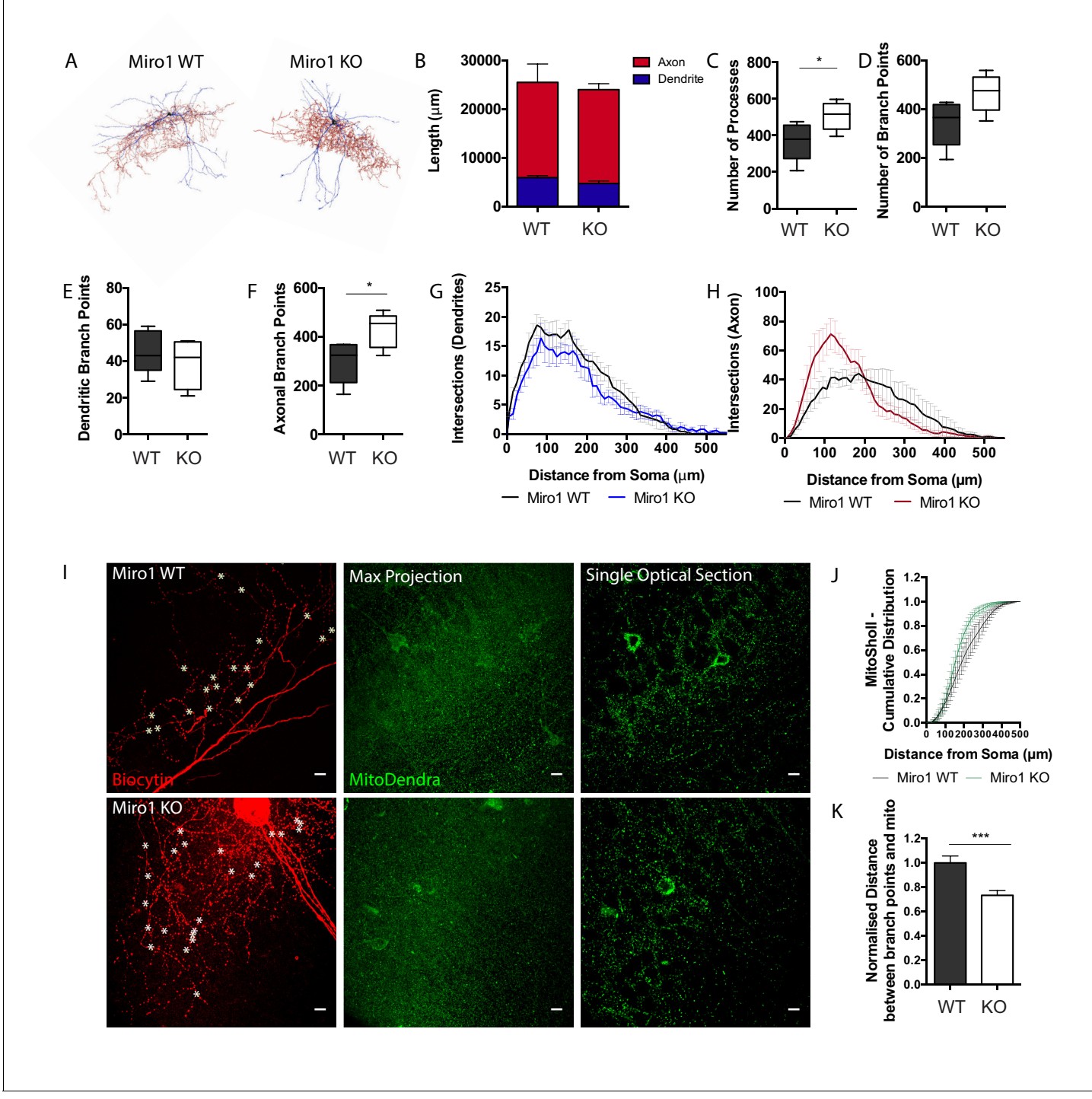

**Figure 3.** Loss of Miro1 results in increased axonal branching in hippocampal parvalbumin interneurons. (A) Max-projected reconstruction of the neurons in *Figure 2D* from Neuromantic software. Dendrites are depicted in blue and axons in red. (B) Quantification of the neuronal length with the dendritic and axonal contribution depicted in blue and red respectively. (C) Boxplot for the quantification of the total number of processes. (D) Boxplot for the quantification of the total number of branch points. (E) Box plot for the quantification of the number of dendritic branch points. (F) Box plot for the quantification of the number of axonal branch points. (G) Number of intersections between dendritic branches and Sholl rings are plotted at distances away from the soma. (H) Number of intersections between axonal branches and Sholl rings are plotted at distances away from the soma. ($n_{WT}$ = five neurons, three slices, two animals, $n_{KO}$ = five neurons, four slices, three animals). (I) Loss of Miro1 results in mitochondria being found closer to points of axonal branching. High-magnification (63×) max-projected confocal stacks of biocytin in a 350 µm hippocampal slice. Analysis was performed in 3D in single optical sections. White stars denote branch points. Scale bar = 10 µm. (J) Cumulative distribution of the MitoDendra+ mitochondrial

*Figure 3 continued on next page*

*Figure 3 continued*

network (MitoSholl) in individual PV+ interneurons ($n_{WT}$ = five neurons, three slices, two animals, $n_{KO}$ = four neurons, four slices, three animals). (K) Bar graph shows the normalized minimum distance between a branch point and a mitochondrion in the confocal stack ($n_{WT}$ = 122 branch points, five neurons, four slices, two animals and $n_{KO}$ = 171 branch points, seven neurons, six slices, three animals).

The online version of this article includes the following source data for figure 3:

**Source data 1.** Source data for PV+ interneuron morphology and mitochondrial distribution.

control and knock-out cells. Sholl analysis revealed a shift in mitochondrial distribution proximal to the soma in Miro1 KO cells (*Figure 3J*). We quantified the minimum normalized distance between branch points and mitochondria (*Figure 3I*) in WT (1 ± 0.06 a.u.) and Miro1 KO (0.7 ± 0.04 a.u.) and found a decrease in the absence of Miro1 (*Figure 3K*, p=0.0003, Mann–Whitney U test). Additionally, the probability of encountering a mitochondrion within 1 µm from the branch point is higher in the Miro1 KO (P = 0.49) when compared to the control condition (P = 0.31) (data not shown). Thus, the loss of Miro1-directed trafficking not only shifted mitochondria away from the presynapse but also placed them closer to locations of axonal branching. The close proximity of mitochondria to axonal branching points in the absence of Miro1 suggests roles for their involvement in driving the formation of axonal segments.

## Loss of Miro1-dependent mitochondrial positioning does not alter parvalbumin interneuron mediated inhibition

We next wanted to address whether the alterations in mitochondrial localization due to impaired trafficking could affect parvalbumin interneuron function and their ability to apply fast perisomatic inhibition in the absence of Miro1. To do that, we recorded the spontaneous inhibitory postsynaptic currents (sIPSCs) that pyramidal cells received in the hippocampus in acute brain slices from WT and Miro1 KO animals (*Figure 4A*). Neither the frequency of the sIPSCs, represented as the inter event interval, nor the amplitude of the responses were different between the two conditions (*Figure 4B, C*), suggesting that the loss of Miro1 did not lead to an alteration in inhibitory synaptic transmission. The recorded sIPSCs however could contain the inhibitory contribution of other interneurons in the network. To specifically assess the PV+ interneuron mediated inhibition, we generated a mouse model where PV+ interneurons could be temporally controlled by the light-induced activation (photoactivation) of channel rhodopsin 2 (ChR2) (*Figure 4D,E*). We then photoactivated PV+ interneurons (1 ms pulse width, 30 repetitions per cell) and quantified the mean evoked IPSCs (eIPSCs) neighbouring pyramidal cells received in the hippocampus (*Figure 4E,F*). We found that there was no change in the amplitude (*Figure 4G*), charge transfer (*Figure 4H*), and decay (*Figure 4I*) between control and Miro1 KO conditions. These data strongly suggest that the PV+ interneuron ability to apply perisomatic inhibition is unaffected by the loss of Miro1 and subsequent changes in mitochondrial dynamics. We also tested whether PV+ cells could sustain inhibition and recover after long-lasting light-activation (*Figure 4J*). We presented a 2 s light train stimulation (40 Hz; 1 ms pulse width) that was repeated 10 times and recorded the eIPSCs from neighbouring pyramidal cells. In order to assess whether the cells recovered in a similar manner after the photo-stimulation, we presented one light pulse at increasing time intervals at the end of every train and measured the eIPSC response. There was no difference in neither the amplitude of the inhibitory currents received by pyramidal cells during the 2 s stimulation (*Figure 4K*) nor in the recovery of post-light train pulses. Thus, Miro1-dependent mitochondrial positioning does not seem to alter the synaptic properties of PV+ cells and may not be important for short-term recovery of the inhibitory responses either, as control and knock-out cells behave similarly (*Figure 4L*). In conclusion, the loss of Miro1 does not seem to alter inhibitory properties of PV+ interneurons in the hippocampus of 2 month old animals.

## Parvalbumin interneurons receive increased excitatory inputs when Miro1 is conditionally knocked-out

Next, we sought to understand whether the loss of Miro1 and subsequent changes in mitochondrial dynamics alter the intrinsic features of PV+ interneurons. We recorded the intrinsic properties of PV + interneurons in WT, HET, and Miro1 KO slices by applying depolarizing and hyperpolarizing pulses (*Figure 4—figure supplement 1A*). We observed no difference in action potential (AP) peak

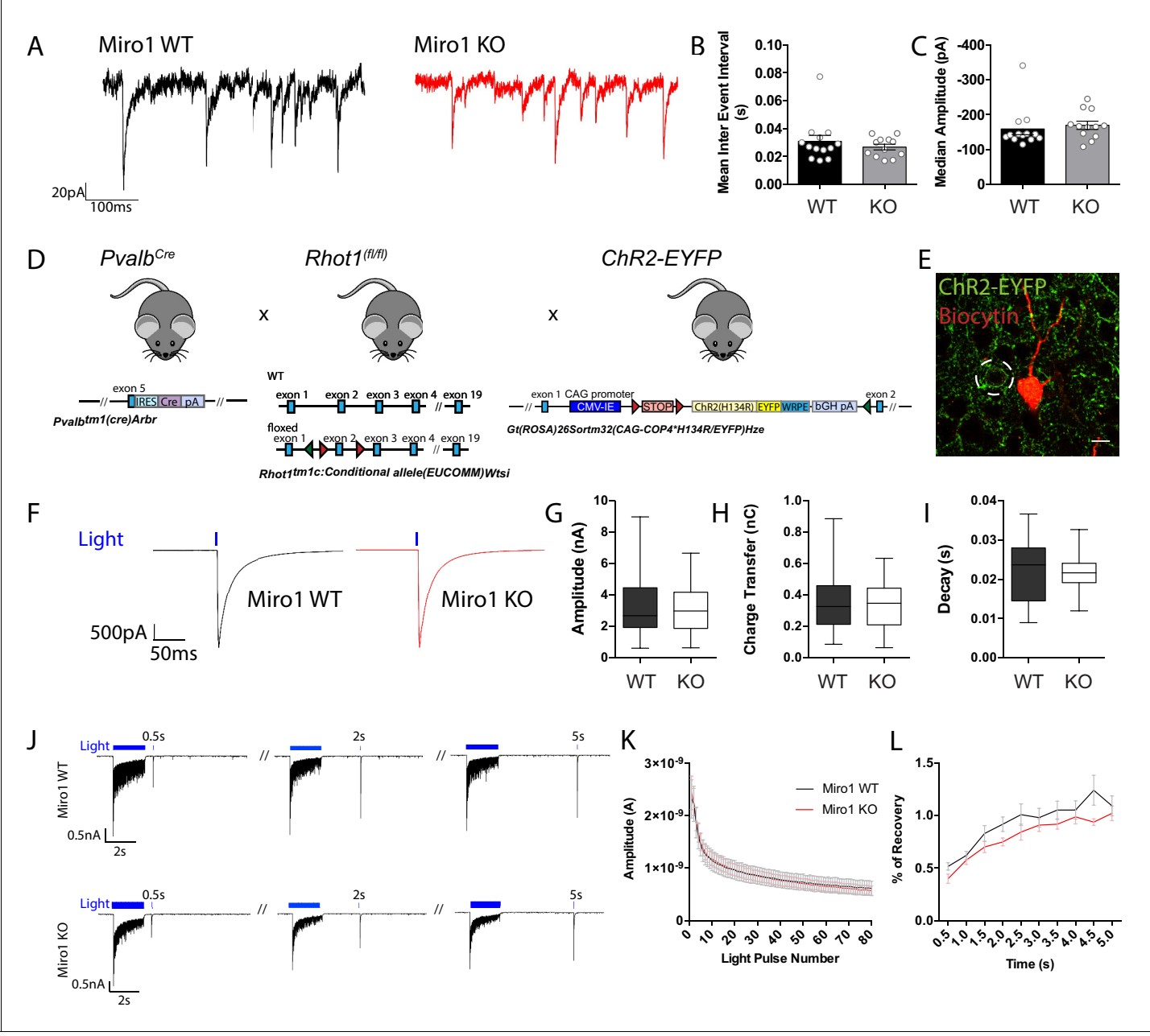

**Figure 4.** Miro1 knock-out does not alter spontaneous and evoked inhibitory synaptic transmission in the hippocampus. (A) Representative electrophysiological traces of spontaneous inhibitory post-synaptic currents from WT (black) and Miro1 KO (red) cells in the hippocampus. (B) Quantification for the mean inter event interval (IEI). (C) Quantification for the median sIPSC amplitude. ($n_{WT}$ = 13 recordings, two animals and $n_{KO}$ = 12 recordings, two animals). (D) Generation of the *Pvalb*$^{Cre}$ *Rhot1* ChR2-EYFP transgenic mouse line. Schematic diagram of the expression of ChR2-EYFP and simultaneous conditional removal of *Rhot1*. When the Cre recombinase is expressed, under the PV+ promoter, the stop-floxed codon is excised from the *Rosa26* locus allowing the downstream expression of ChR2-EYFP. Additionally, the second exon of the *Rhot1* gene is found between two loxP sites and also removed selectively in PV+ interneurons. (E) Example of confocal image from a biocytin-filled recorded pyramidal cell in the hippocampus (red) in close proximity to an EYFP+ PV+ interneuron (green). Scale bar = 10 μm. (F) Representative traces from light evoked inhibitory postsynaptic current (eIPSC) in WT (black) and Miro1 KO (red) cells in acute brain slices ($n_{WT}$ = 23 recordings, four animals and $n_{KO}$ = 23 recordings, four animals). (G) Boxplot for the quantification of peak amplitude. (H) Boxplot for the quantification of charge transfer. (I) Boxplot for the quantification of decay. (J) Control and conditional knock-out cells can sustain inhibition and recover with similar rates after long-lasting photostimulation. Example traces from the inhibitory responses pyramidal cells received in WT and Miro1 KO slices during light train stimulation (40 Hz for 2 s; 1 ms pulse width). (K) Mean amplitude of each peak during the light train stimulation ($n_{WT}$ = 21 recordings, four animals and $n_{KO}$ = 24 recordings, four animals). (L) Quantification of the percentage recovery after light stimulation of all cells at increasing time intervals from the end of the light train ($n_{WT}$ = 21 recordings, four animals and $n_{KO}$ = 18 recordings, three animals).

*Figure 4 continued on next page*

*Figure 4 continued*

The online version of this article includes the following source data and figure supplement(s) for figure 4:

**Source data 1.** Source data for PV+ interneuron function.
**Figure supplement 1.** Intrinsic properties of PV+ interneurons in the absence of Miro1.
**Figure supplement 1—source data 1.** Source data for PV+ interneuron intrinsic properties.

amplitude (*Figure 4—figure supplement 1B*), input resistance (*Figure 4—figure supplement 1E*), threshold to fire (*Figure 4—figure supplement 1F*), and spike rate at 40 pA above rheobase current (*Figure 4—figure supplement 1D*) in the three conditions. The AP half-width was slightly higher in WT (0.4 ± 0.03 ms) when compared to HET (0.3 ± 0.02 ms) and Miro1 KO (0.3 ± 0.02 ms) recordings (*Figure 4—figure supplement 1C*, $p_{HET}$ = 0.048, $p_{KO}$ = 0.864, Tukey's multiple comparisons test, ordinary one-way ANOVA F(2, 36) = 3.179, P = 0.0535). The membrane time constant was slightly decreased in HET (8 ± 0.8 ms) and Miro1 KO (8 ± 0.6 ms) animals when compared to control (12 ± 2.0 ms, *Figure 4—figure supplement 1G*, $p_{HET}$ = 0.073, $p_{KO}$ = 0.046, Tukey's multiple comparisons test, ordinary one-way ANOVA F(2, 36) = 3.391, P = 0.0448). We speculate that although the differences in intrinsic properties were small, the loss of Miro1 from PV+ interneurons might potentially result in shorter integration time and faster response to excitatory inputs. The change in membrane time constant could also potentially reflect the changes to the morphology of the cell (*Isokawa, 1997*).

To investigate whether PV+ interneurons received differential synaptic inputs, we performed whole-cell recordings and acquired the spontaneous excitatory postsynaptic currents (sEPSCs) in the hippocampus of acute brain slices (*Figure 5A*). The sEPSC frequency was significantly increased, as shown by the decrease in the mean inter event interval (IEI) between WT (0.03 ± 0.009 s) and Miro1 KO (0.02 ± 0.003 s) cells (*Figure 5B*, p = 0.025, Mann–Whitney U test). Additionally, the amplitude of sEPSCs was elevated in Miro1 KO (−188 ± 17.2 pA) slices when compared to control slices (−130 ± 7.8 pA, *Figure 5C*, p = 0.009, unpaired t-test with Welch's correction). Thus, the increase in sEPSC frequency and amplitude suggest that PV+ interneurons receive an enhanced excitatory drive when Miro1 was knocked-out. To address whether the increase in excitation was specific to PV+ interneurons or a general enhancement in excitatory tone, we performed immunohistochemistry experiments on fixed brain tissue of WT and Miro1 KO using antibodies against the pre-synaptic marker Bassoon and the post-synaptic marker Homer (*Figure 5—figure supplement 1A*). The total immuno-stained area that each synaptic marker occupied, and the extent of the Bassoon–Homer overlap (colocalization) did not differ in the two conditions, suggesting that the loss of Miro1 from PV+ interneurons does not change the levels of excitatory synapses in the hippocampus. Next, we performed extracellular recordings of field excitatory postsynaptic potentials (fEPSP) at increasing electrical stimulations (*Figure 5—figure supplement 1B*). The input–output (I/O) relationship indicated that basal synaptic transmission was increased in Miro1 KO compared to WT, suggesting an increase in the overall excitatory tone. Thus, excitatory transmission may be potentiated in the absence of Miro1 (0.19 ± 0.033 a.u.), compared to the control (0.10 ± 0.005 a.u.), as indicated by the increase in the fEPSP slope at high stimulations (*Figure 5—figure supplement 1B*, p = 0.0018, Tukey's multiple comparisons test, ordinary one-way ANOVA F (3, 139) = 17.58, P < 0.0001). These data suggest that network excitation is enhanced in the hippocampus when Miro1 is knocked-out from PV+ interneurons.

## Alterations in Miro1-dependent mitochondrial positioning change hippocampal network activity and anxiety-related animal behavior

We then sought to examine whether the Miro1-dependent alterations in mitochondrial distribution could result in altered neuronal network activity. Given that γ-oscillations are energetically very costly (*Kann et al., 2014*) and depend on proper PV+ interneuron function, we wanted to see whether the loss of Miro1 could have an impact on the ability of PV+ cells to synchronize neuronal networks. We measured carbachol induced γ-oscillations in the CA3 area of the hippocampus by recording local field potentials in acute brain slices of WT, HET, and Miro1 KO animals (*Figure 5D,E*). Even though the peak power and power area were not significantly different (*Figure 5G,H*), we found a small but significant increase in the peak frequency of the oscillations that appeared to be gene-dose dependent between the different genotypes (WT 31 ± 0.6 Hz, HET 35 ± 0.6 Hz, Miro1 KO 37 ± 0.9 Hz,

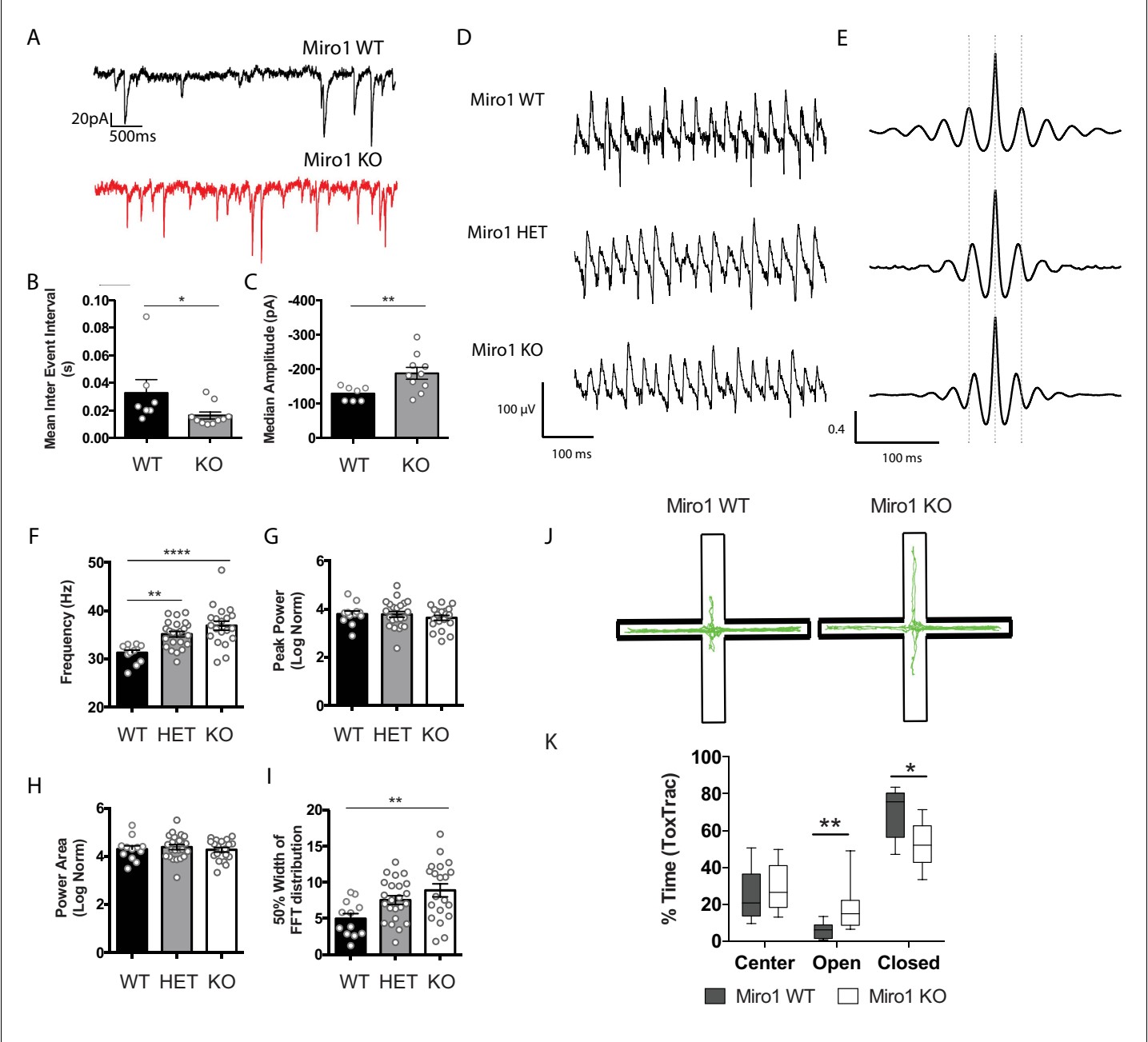

**Figure 5.** Miro1 knock-out results in altered hippocampal network activity and anxiety-related behavior. (**A**) Miro1 KO PV+ interneurons received increased glutamateric input. Representative electrophysiological traces from WT (black) and Miro1 KO (red) cells. (**B**) Quantification for the mean inter event interval (IEI). (**C**) Quantification for the median amplitude ($n_{WT}$ = seven recordings, two animals and $n_{KO}$ = 10 recordings, two animals). (**D**) Loss of Miro1 increases the frequency of γ-oscillations. Representative local field potential recordings from the stratum pyramidale of the CA3 hippocampal area in acute brain slices from WT, HET, and Miro1 KO mice. (**E**) Representative auto-correlogram of γ-oscillations from WT, HET, and Miro1 KO animals. (**F**) Quantification of the peak frequency. (**G**) Quantification of the normalized peak power. (**H**) Quantification of the normalized power area. (**I**) Quantification of the 50% width dispersion of the FFT distribution ($n_{WT}$ = 12 slices, two animals, $n_{HET}$ = 23 slices, six animals, and $n_{KO}$ = 20 slices, six animals). (**J**) Assessment of anxiety-related behavior using the elevated plus maze (EPM). Schematic diagram of the EPM and representative ToxTrac trajectories (green) from WT and Miro1 KO animals. (**K**) Boxplot for the quantification of the percentage of time that WT and Miro1 KO animals spent in the closed, open arms and center of the EPM ($n_{WT}$ = nine animals, $n_{KO}$ = eight animals).

The online version of this article includes the following source data and figure supplement(s) for figure 5:

**Source data 1.** Source data for excitatory drive, network activity and anxiety-related behavior.

**Figure supplement 1.** Loss of Miro1 is accompanied by a general increase in excitatory drive in the hippocampus without altering the total levels of excitatory pre- and post- synaptic markers in the hippocampus.

*Figure 5 continued on next page*

*Figure 5 continued*

**Figure supplement 1—source data 1.** Source data for excitation in hippocampus.
**Figure supplement 2.** Loss of Miro1 does not affect husbandry behavior, motor coordination, short-term memory, and spatial exploration.
**Figure supplement 2—source data 1.** Source data for animal behavior experiments.

*Figure 5F*, $p_{HET}$ = 0.004, $p_{KO}$ < 0.0001, Tukey's multiple comparisons test, ordinary one-way ANOVA F(2, 52) = 11.58, P < 0.0001). We then quantified the 50% width of the power spectrum distribution as a measure of the oscillation variability (*Figure 5I*). We noticed that the distribution was significantly broader in HET (8 ± 0.6) and Miro1 KO (9 ± 0.9) when compared to control (5 ± 0.7), suggesting increased variability of the γ-cycle duration (*Figure 5I*, $p_{KO}$ = 0.005, Tukey's multiple comparisons test, ordinary one-way ANOVA F(2, 52) = 5.407, P = 0.0074). These data demonstrate that the loss of Miro1 and subsequent changes in mitochondrial trafficking and distribution in PV+ interneurons are sufficient to alter hippocampal network activity, further involving mitochondria in the modulation of ex vivo γ-oscillations (*Inan et al., 2016*; *Kann, 2011*; *Kann et al., 2011*).

Finally, we wanted to see whether the loss of Miro1 from PV+ interneurons was sufficient to induce functional changes in the behavior of these animals. We observed no changes in husbandry behavior (*Figure 5—figure supplement 2A*) assessed by the shredding of Nestlets (*Deacon, 2006*), motor coordination and learning (*Figure 5—figure supplement 2B*) on the rotarod (*Deacon, 2013*), and short-term memory (*Figure 5—figure supplement 2C*) in the T-Maze (*Deacon and Rawlins, 2006*). We also did not observe a robust difference in spatial exploration of WT and Miro1 KO animals in the open-field assessment (*Figure 5—figure supplement 2D*; *Seibenhener and Wooten, 2015*). The velocity (*Figure 5—figure supplement 2E*), distance traveled (*Figure 5—figure supplement 2F*), and time spent in the areas of the arena (*Figure 5—figure supplement 2G*) were not statistically different between the two genotypes. Finally, we tested anxiety-like behavior of littermate control and conditional knock-out animals in the elevated plus maze (EPM) (*Figure 5J*). As expected, both WT and Miro1 KO mice spent the majority of time in the closed arms due to the natural aversive behavior to light. The Miro1 KO animals however spent more time in the open arms (18 ± 5%) than their littermate controls (6 ± 1%, *Figure 5K*, p = 0.004, Mann–Whitney U test). The performance of Miro1 KO animals in the EPM suggests that they may exhibit a reduced anxiety-like phenotype. In conclusion, the selective loss of Miro1 from PV+ interneurons results in changes in mitochondrial dynamics, axonal remodeling, and network activity that might give rise to anxiety related phenotypes, affecting the behavior of these animals.

## Discussion

In this study, we demonstrate that the conditional removal of Miro1 in PV+ cells resulted in a mitochondrial trafficking impairment. Furthermore, the loss of Miro1 led to an accumulation of mitochondria in the soma of PV+ cells and their depletion from axonal presynaptic terminals. The relocation of mitochondria closer to sites of axonal branching was associated with a selective enrichment in axonal branches proximal to the cell body. Interestingly, these animals also exhibited increased network excitation, altered γ-oscillations, and a reduced anxiety-like phenotype. Our data suggest that Miro1-dependent mitochondrial positioning is implicated in shaping hippocampal network activity and animal behavior.

The enrichment of mitochondria in PV+ interneurons denotes their important role in the correct function of these neurons (*Gulyás et al., 2006*; *Inan et al., 2016*; *Lin-Hendel et al., 2016*; *Paul et al., 2017*). Our immunohistochemical data also demonstrate that PV+ interneurons display increased levels of the subunit IV of cytochrome c oxidase (COX-IV) (*Figure 1A*). This is consistent with evidence proposing high levels of oxidative phosphorylation proteins such as complex I, cytochrome c oxidase, cytochrome c, and ATP synthase in these cells (*Gulyás et al., 2006*; *Kann et al., 2011*; *Paul et al., 2017*). This observation could further support the notion that PV+ interneurons are metabolically more active to meet their high-energy demands and suggests that stringent dependence on mitochondrial function could render PV+ interneurons susceptible to incidents of mitochondrial damage.

The generation of the *Pvalb^Cre Rhot1*^flox/flox MitoDendra transgenic mouse permitted the visualization of mitochondria and the conditional removal of Miro1 specifically in PV+ cells (*Figure 1—figure supplement 1*). Given the role of the adaptor protein Miro1 in the bidirectional trafficking of mitochondria to locations of high-energy demand (*Birsa et al., 2013*; *Guo et al., 2005*; *López-Doménech et al., 2016*; *López-Doménech et al., 2018*; *Macaskill et al., 2009*; *Nguyen et al., 2014*; *Saotome et al., 2008*; *Wang and Schwarz, 2009*), we examined the importance of correct mitochondrial transport in PV+ interneurons. The conditional removal of Miro1 resulted in a decrease in mitochondrial trafficking, consistent with the functional role of Miro1 (*Figure 1B*). A small subset of mitochondria remained mobile in the absence of Miro1, indicating that mitochondrial transport was not completely abolished in Miro1 KO. Other studies also report a reduction, but not complete cessation of mitochondrial transport, upon Miro1 loss (*López-Doménech et al., 2016*; *Macaskill et al., 2009*; *Russo et al., 2009*), hinting to the existence of alternative mitochondrial transport mechanisms. The physical attachment of mitochondria to other motor–adaptor proteins might still facilitate organelle transport. Our lab has demonstrated that Trak1/2 motors can still be recruited to the mitochondria in mouse embryonic fibroblasts (MEFs), in a Miro1-independent fashion (*López-Doménech et al., 2018*). The mobile mitochondria in PV+ interneurons exhibited shorter traveled trajectories in Miro1 KO (*Figure 1E*). Thus, it is also possible that the remaining moving mitochondria engaged with cytoskeletal elements other than microtubules. Indeed, involvement of the actin cytoskeleton has been implicated in short-range mitochondrial movement (*Chada and Hollenbeck, 2004*; *Morris and Hollenbeck, 1995*). Still, the remaining moving mitochondria in our model exhibited directional, rather than random movement, which favors transport along microtubules.

The loss of Miro1 from PV+ interneurons resulted in mitochondrial clustering in the cell bodies, suggesting that mitochondria are no longer able to exit the soma due to impaired mitochondrial trafficking (*Figure 2A*). Similar perinuclear clustering has also been reported upon the loss of Miro1 in MEFs (*López-Doménech et al., 2018*; *Nguyen et al., 2014*) and in the *Camk2a^Cre Rhot1*^flox/flox mouse at 4 months of age (*López-Doménech et al., 2016*). The loss of Miro1-directed mitochondrial transport was accompanied by a redistribution in the mitochondrial network in PV+ interneurons. While the majority of PV+ axonal boutons contain mitochondria (*Figure 2G*), loss of Miro1 resulted in their depletion from these sites (*Figure 2F*), suggesting a role for Miro1 in mitochondrial capture in presynaptic terminals along the PV+ interneuron axon. Mutations in dMiro, the *Drosophila* orthologue, impair mitochondrial trafficking and deplete mitochondria from the axon, altering the morphology of both the axon and synaptic boutons in the neuromuscular junction (*Guo et al., 2005*). Miro1 also mediates the activity-dependent repositioning of mitochondria to synapses in excitatory neurons (*Vaccaro et al., 2017*). Collectively, this evidence proposes a role for Miro1-directed mitochondrial trafficking in the fine tuning of presynaptic mitochondrial occupancy. We speculate that the loss of mitochondria from axonal terminals in Miro1 KO cells could be the consequence of defective protein-protein interactions with the tethering machinery, resulting in the inability of mitochondria to be retained in the presynaptic space. Mitochondrial transport on actin filaments mediates local distribution along the axon and destabilization of F-actin reduces mitochondrial docking (*Chada and Hollenbeck, 2003*; *Chada and Hollenbeck, 2004*; *Hirokawa et al., 2010*; *Shlevkov et al., 2019*; *Smith and Gallo, 2018*). It is therefore possible that the loss of Miro1 might have disrupted the interaction between mitochondria and the actin cytoskeleton resulting in reduced presynaptic capture for example due to impairments in the Myo19-dependent coupling to actin in the absence of Miro1 (*López-Doménech et al., 2018*).

Neuronal mitochondrial misplacement is thought to impact on neuronal morphology as mitochondria are no longer precisely located in places where they are actively required to provide energy and buffer $Ca^{2+}$ (*Devine and Kittler, 2018*; *Guo et al., 2005*; *Liu and Shio, 2008*; *López-Doménech et al., 2016*). By reconstructing the morphology of PV+ interneurons in brain slices (*Figure 3A*), we reported a selective enrichment in axonal, but not dendritic branching, proximal to the cell soma upon loss of Miro1 (*Figure 3H*). Interestingly, the loss of Miro1-directed mitochondrial trafficking and depletion of mitochondria from axonal boutons were accompanied by a general shift in the mitochondrial distribution closer to the soma (*Figure 3J*). Since mitochondria were found closer to sites of branch points in the absence of Miro1, our data suggest that the enhancement in axonal branching could be attributed to the perturbed mitochondrial distribution (*Figure 3I*). Mitochondrial arrest and presynaptic capture is necessary for axon extension and branching formation, as

mitochondria can locally provide energy (*Courchet et al., 2013*; *Sainath et al., 2017*; *Smith and Gallo, 2018*; *Spillane et al., 2013*). The extension and retraction of the axon are developmentally dynamic processes (*Chattopadhyaya et al., 2004*; *Chattopadhyaya et al., 2007*; *Huang et al., 2007*) and the Miro1-dependent changes in mitochondrial distribution could promote either the formation of branches close to the cell body or the reorganization of existing branches.

We then examined whether the spatiotemporal regulation of mitochondrial positioning could impact on PV+ interneuron function and network activity. PV+ interneurons are critical elements in generating and maintaining γ-oscillations (*Antonoudiou et al., 2020*; *Bartos et al., 2007*; *Buzsáki and Wang, 2012*; *Cardin et al., 2009*; *Sohal et al., 2009*), and these oscillations heavily rely on intact mitochondrial function (*Galow et al., 2014*; *Huchzermeyer et al., 2008*; *Huchzermeyer et al., 2013*; *Kann et al., 2011*; *Kann et al., 2016*; *Whittaker et al., 2011*). Indeed, pharmacological inhibition of complex I with rotenone, and uncoupling of oxidative phosphorylation with FCCP abolishes the power of γ-oscillations (*Whittaker et al., 2011*), while rotenone also reduced oxygen consumption in the CA3 area of the hippocampus, coupling respiration to network activity (*Kann et al., 2011*). By recording carbachol-induced rhythmic network activity in the CA3 area of the hippocampus, we reported a mild increase in the frequency and variability of γ-oscillations in a Miro1 dose-dependent manner (*Figure 5D,E*). These results suggest that in addition to the proper functioning of mitochondria, their precise positioning along the axon might be important in the fine tuning of rhythmic oscillations in γ-frequency band. The width of the power spectra was wider when Miro1 was knocked-out from PV+ interneurons (*Figure 5I*), suggesting higher variability in the duration of γ-cycles. Surprisingly, the Miro1-directed changes in mitochondrial dynamics were not sufficient to compromise PV+ interneuron output (*Figure 4*), which would be directly reflected on the generation and maintenance of γ-oscillations in the hippocampus. Alterations in axonal branching could have an implication on the innervation of postsynaptic targets (*Huang et al., 2007*), and it is therefore possible that the precise PV+ interneuron innervation of pyramidal cells changed (*Figure 3H*), resulting in an increase in the excitation PV+ interneurons received in the hippocampus (*Figure 5A,B*), without compromising inhibition (*Figure 4*). We speculate that the extent of the axon reach in the hippocampus might be reduced resulting in each PV+ interneuron inhibiting a different number of postsynaptic cells in the local network, introducing variability in the synchronization of cell assemblies and an overall enhancement in the excitatory tone in the hippocampus (*Figure 5—figure supplement 1B*). Furthermore, recent work proposed that neuronal entrainment at γ-frequencies during development can impact neuronal morphology of cortical cells (*Bitzenhofer et al., 2019*). Thus, it will be of great interest to explore the relationship between network activity at γ-frequency and neuronal morphology.

Surprisingly, given the established role of mitochondria in presynaptic release (*Devine and Kittler, 2018*; *Sun et al., 2013*), neither spontaneous inhibitory postsynaptic transmission nor PV+ interneuron-mediated evoked inhibition was altered by the reduction of mitochondria in axonal boutons in Miro1 KO (*Figure 4*), demonstrating that the changes in mitochondrial distribution associated with the impairments in Miro1-directed trafficking are not sufficient to alter the perisomatic inhibition applied by PV+ interneurons. The ability of PV+ interneurons to sustain inhibition and recover remained intact in the absence of Miro1 (*Figure 4*). The $Ca^{2+}$ buffering capabilities of the parvalbumin protein might compensate for the absence of the local mitochondrial pool and participate in synaptic release by sequestering $Ca^{2+}$. Genetic removal of the parvalbumin protein has an impact on inhibitory transmission, synaptic facilitation, and network activity (*Caillard et al., 2000*; *Eggermann and Jonas, 2011*; *Orduz et al., 2013*; *Vreugdenhil et al., 2003*). Thus, $Ca^{2+}$ signaling in PV+ interneurons may be modulated by the interplay of various buffers such as mitochondria and parvalbumin itself. It is also possible that the increased cellular concentration of mitochondria in PV+ interneurons ensures sufficient metabolic capacity to support their energetic demands. Although reports have shown the requirement of local ATP synthesis and provision during sustained synaptic activity (*Rangaraju et al., 2014*; *Sun et al., 2013*), which is facilitated via the $Ca^{2+}$ uptake by mitochondria (*Ashrafi et al., 2020*), the diffusion of ATP from mitochondria rich regions to boutons devoid of mitochondria may be sufficient to sustain the energetic requirements of synapses (*Pathak et al., 2015*). Moreover, the activity-driven glycolysis via glucose transporter mobilization at presynaptic terminals (*Ashrafi et al., 2017*) may also provide the necessary energy to sustain PV+ interneuron mediated inhibition during synaptic release. Further investigation is required to

understand how the changes in Miro1-dependent mitochondrial positioning and enhanced axonal branching could be implicated in varying the strength and pattern of inhibition in the hippocampus.

Changes in PV+ interneuron excitability and network activity have been associated with alterations in behavior and emergence of neurological and neuropsychiatric disorders (*Inan et al., 2016*; *Marín, 2012*; *Pelkey et al., 2017*; *Zou et al., 2016*). Thus, we performed experiments to test exploratory locomotion, memory, and anxiety in control and Miro1 KO mice and assess whether the alterations in mitochondrial dynamics and axonal morphology were physiologically relevant by having an impact on behavior. Miro1 KO animals presented no deficits in husbandry behavior, motor coordination and learning, short-term memory, and spatial exploration when compared to their littermate controls (*Figure 5—figure supplement 2*). Interestingly, Miro1 KO animals exhibited anxiolytic behavior, demonstrated as increased time spent in the open arms of the elevated plus maze (EPM). This observation raises the interesting question about the role of Miro1-dependent mitochondrial dynamics and PV+ interneuron signaling in distinct brain areas involved in emotional behaviors such as the ventral hippocampus, amygdala, and prefrontal cortex (*Janak and Tye, 2015*). In conclusion, our findings demonstrate that the Miro1-directed spatiotemporal positioning of mitochondria in PV+ interneurons can modulate axon morphology, the frequency of hippocampal network oscillations at the γ-band range and potentially influence stress and emotional behaviors.

# Materials and methods

## Key resources table

| Reagent type (species) or resource | Designation | Source or reference | Identifiers | Additional information |
|---|---|---|---|---|
| Gene (mouse) | *Rhot1* | GenBank | Gene ID: 59040 | |
| Strain, strain background (include species and sex here) | Mouse B6;129S—Gt(ROSA)26Sortm1(CAG—COX8A/Dendra2)Dcc/J | The Jackson Laboratory | Stock number 018385 | |
| Strain, strain background (include species and sex here) | Mouse: Rhot1tm1a(EUCOMM)Wtsi | Wellcome Trust Sanger Institute | MBTN EPD0066 2 F01 | |
| Strain, strain background (include species and sex here) | Mouse: B6;129S—Gt(ROSA)26Sortm32(CAG—COP4*H134R/EYFP)Hze/J | The Jackson Laboratory | Stock number 012569 | |
| Biological sample (include species here) | Mouse Organotypic Brain Slices | This Paper | | |
| Biological sample (include species here) | Mouse Acute Brain Slices | This Paper | | |
| Antibody | Anti-Parvalbumin (mouse monoclonal) | Millipore | Cat# MAB1572 RRID:AB_2174013 | 1:500 |
| Antibody | Anti-Rhot1 (Miro1) (rabbit polyclonal) | Atlas | Cat# HPA010687 RRID:AB_1079813 | 1:100 |
| Antibody | Anti-COX-IV (rabbit polyclonal) | Abcam | Cat# Ab16056 RRID:AB_443304 | 1:500 |
| Antibody | Anti-Bassoon (mouse monoclonal) | Abcam | Cat# ab82958 RRID:AB_1860018 | 1:500 |
| Antibody | Homer (rabbit polyclonal) | Synaptic Systems | Cat# 160 002 RRID:AB_2120990 | 1:500 |
| Antibody | Donkey Anti-Mouse Alexa Fluor 488 | Jackson ImmunoResearch | Cat# 715-545-151 RRID:AB_2341099 | 1:500-1:1000 |
| Antibody | Goat Anti-Rabbit Alexa Fluor 555 | Thermo Fisher Scientific | Cat# A-21430 RRID:AB_2535851 | 1:500-1:1000 |

*Continued on next page*

*Continued*

| Reagent type (species) or resource | Designation | Source or reference | Identifiers | Additional information |
|---|---|---|---|---|
| Antibody | Donkey Anti-Rabbit Alexa Fluor 568 | Thermo Fisher Scientific | Cat# A-10042 RRID:AB_2534017 | 1:500-1:1000 |
| Antibody | AffiniPure Fab Fragment Goat Anti-Mouse IgG (H+L) | Jackson ImmunoResearch | Cat# 115-007-003 RRID:AB_2338476 | 50 µg/ml |
| Sequence-based reagent | Rhot1 Forward: Rhot1_16_F | This paper, *López-Doménech et al., 2016*, Sigma-Aldrich | PCR Primers | TTAGGATTTGTACTTTGCCCCTG |
| Sequence-based reagent | Rhot1 Reverse: Rhot1_16_R | This paper, *López-Doménech et al., 2016*, Sigma-Aldrich | PCR Primers | AAAACCCTTCCTGCATCACC |
| Sequence-based reagent | Cas | This paper, *López-Doménech et al., 2016*, Sigma-Aldrich | PCR Primers | TCGTGGTATCGTTATGCGCC |
| Sequence-based reagent | MitDend WT Forward | This paper, Sigma-Aldrich | PCR Primers | CCAAAGTCGCTCTGAGTTGTTATC |
| Sequence-based reagent | MitDend WT Reverse | This paper, Sigma-Aldrich | PCR Primers | GAGCGGGAGAAATGGATATG |
| Sequence-based reagent | MitDend Mut Reverse | This paper, Sigma-Aldrich | PCR Primers | TCAATGGGCGGGGGTCGTT |
| Sequence-based reagent | Cre Forward | This paper, Sigma-Aldrich | PCR Primers | CACCCTGTTACGTATAGCCG |
| Sequence-based reagent | Cre Reverse | This paper, Sigma-Aldrich | PCR Primers | GAGTCATCCTTAGCGCCGTA |
| Sequence-based reagent | LacZ_2_small_F | This paper, *López-Doménech et al., 2016*, Sigma-Aldrich | PCR Primers | ATCACGACGCGCTGTATC |
| Sequence-based reagent | LacZ_2_small_R | This paper, *López-Doménech et al., 2016*, Sigma-Aldrich | PCR Primers | ACATCGGGCAAATAATATCG |
| Sequence-based reagent | ChR Forward | This paper, Sigma-Aldrich | PCR Primers | AAGGGAGCTGCAGTGGAGTA |
| Sequence-based reagent | ChR Reverse | This paper, Sigma-Aldrich | PCR Primers | CCGAAAATCTGTGGGAAGTC |
| Sequence-based reagent | ChR mut Forward | This paper, Sigma-Aldrich | PCR Primers | ACATGGTCCTGCTGGAGTTC |
| Sequence-based reagent | ChR mut Reverse | This paper, Sigma-Aldrich | PCR Primers | GGCATTAAAGCAGCGTATCC |
| Peptide, recombinant protein | Streptavidin conjugated to Alexa Fluor 555 | Invitrogen | S32355 | 1:500 |
| Chemical compound, drug | Carbachol | Sigma Aldrich | 51.83.2 | [5 µM] |
| Chemical compound, drug | Biocytin | Sigma Aldrich | B4261 | [3–4 mg/ml] |
| Software, algorithm | Fiji | *Schindelin et al., 2012* | https://www.fiji.sc/ | |
| Software, algorithm | Simple Neurite Tracer | *Longair et al., 2011* | https://imagej.net/SNT | |
| Software, algorithm | Neuromantic | *Myatt et al., 2012* | https://www.reading.ac.uk/neuromantic/body_index.php | |
| Software, algorithm | ToxTrac | *Rodriguez et al., 2018* | https://sourceforge.net/projects/toxtrac/ | |
| Software, algorithm | Matlab_R2015a | Mathworks | https://www.mathworks.com/products/matlab.html | |
| Software, algorithm | Prism 6 | Graphpad | https://www.graphpad.com/scientific-software/prism/ | |

*Continued on next page*

*Continued*

| Reagent type (species) or resource | Designation | Source or reference | Identifiers | Additional information |
|---|---|---|---|---|
| Software, algorithm | IgorPro 6.3 | Wavemetrics | https://www.wavemetrics.com/ | |
| Other | DAKO mounting media | Agilent Technologies | S3023 | |
| Other | Omnipore membrane inserts | Millipore | Cat no. FHLC01300 | |

## Experimental model

### Animals

All experimental procedures were carried out in accordance with institutional animal welfare guidelines and licensed by the UK Home Office in accordance with the Animals (Scientific Procedures) Act 1986. Animals were maintained under controlled conditions (temperature 20 ± 2℃; 12 hr light–dark cycle). Animals of either sex were used for all experiments. The $Pvalb^{Cre}$ line (Stock Number 008069) has been previously described in *Hippenmeyer et al., 2005*. The *Rhot1* transgenic line (Rhot1tm1a (EUCOMM)Wtsi) was obtained from the Wellcome Trust Sanger Institute (MBTN EPD0066 2 F01), and the floxed mouse has been previously generated using the Flp recombination strategy described here (*López-Doménech et al., 2016*). Briefly, the exon 2 of *Rhot1* gene (chromosome 11) is flanked by two LoxP sites and Cre recombination results in the deletion of exon 2. The stop-floxed MitoDendra line (*B6; 129S Gt(ROSA)26Sortm*1(*CAGCOX*8*A/Dendra*2)*Dcc/J*) (Stock number 018385) and the stop-floxed ChR2-EYFP (*B6;129SGt(ROSA)26Sortm*32(*CAGCOP*4*H*134*R/EYFP)Hze/J*) (Stock number 012569) were also obtained from The Jackson Laboratory. The MitoDendra line has been previously described in *Pham et al., 2012* and the ChR2-EYFP line in *Madisen et al., 2012*. For experiments where the expression of a fluorescent reporter was not necessary, (electrophysiology and behavior) we crossed Miro1 floxed animals ($Rhot1^{flox/flox}$) with $Pvalb^{Cre+/-}$ $Rhot1^{flox/flox}$ to generate animals that contained at least one allele of the Cre recombinase under the control of the PV+ promoter. From these crosses 50% of the littermates were Cre positive (conditional knock-out) and 50% were Cre negative (controls). For experiments implicating the expression of the MitoDendra reporter, we generated crosses with animals that were $Pvalb^{Cre+/+}$; $Rhot1^{flox/+}$ and $MitoDendra^{+/+}$ with $Pvalb^{Cre+/+}$; $Rhot1^{flox/flox}$ and $MitoDendra^{+/+}$ animals. Therefore, 50% of the progeny constitute conditional knock-out (Miro1 KO) animals while the remaining 50% are control (Miro1 HET) animals. A similar strategy was followed to produce controls and conditional Miro1 knock-out and expressing ChR2-YFP in PV+ interneurons.

### Genotyping

Genotyping was carried out following Sanger recommended procedures on ear biopsies for adult mice and tail biopsies for neonatal mice (<P10). PCRs were performed using the genotyping primers provided in *Table 1*.

## Hippocampal brain slice preparations

### Acute brain slices

For patch clamp electrophysiology and oscillation experiments, adult mice (>P60) were anesthetized using 4% isoflurane followed by decapitation, and the brains were extracted in warm (30–35℃) sucrose solution (40 mM NaCl, 3 mM KCl, 7.4 mM $MgSO_4.7H_2O$, 150 mM sucrose, 1 mM $CaCl_2$, 1.25 mM $NaH_2PO_4$, 25 mM $NaHCO_3$, and 15 mM glucose; osmolality 300 ± 10 mOsmol/kg). Horizontal hippocampal slices (350 μm thick) were cut using a vibratome (Leica VT1200S) and were placed in an interface chamber containing warm artificial cerebrospinal fluid (aCSF) (126 mM NaCl, 3.5 mM KCl, 2 mM $MgSO_4.7H_2O$, 1.25 mM $NaH_2PO_4$, 24 mM $NaHCO_3$, 2 mM $CaCl_2$, and 10 mM glucose; osmolality 300 ± 10 mOsmol/kg). For the fEPSP experiments, adult mice were sacrificed by cervical dislocation followed by decapitation and brains were extracted in ice-cold dissecting solution [87 mM NaCl, 25 mM $NaHCO_3$, 10 mM glucose, 75 mM sucrose, 2.5 mM KCl, 1.25 mM $NaH_2PO_4$, 0.5 mM $CaCl_2$, and 7 mM $MgCl_2$]. Transverse hippocampal slices (300 μm thick) were cut using a vibratome (Leica, VT-1200S) and stored in dissecting solution at 35℃ for 30 min and then in

**Table 1.** Sequencing primers.

| Genotyping primer | Sequence (5' to 3') |
| --- | --- |
| Rhot1 forward: Rhot1_16_F | TTAGGATTTGTACTTTGCCCCTG |
| Rhot1 reverse: Rhot1_16_R | AAAACCCTTCCTGCATCACC |
| Cas | TCGTGGTATCGTTATGCGCC |
| MitDend WT forward | CCAAAGTCGCTCTGAGTTGTTATC |
| MitDend WT reverse | GAGCGGGAGAAATGGATATG |
| MitDend Mut reverse | TCAATGGGCGGGGGTCGTT |
| Cre forward | CACCCTGTTACGTATAGCCG |
| Cre reverse | GAGTCATCCTTAGCGCCGTA |
| LacZ_2_small_F | ATCACGACGCGCTGTATC |
| LacZ_2_small_R | ACATCGGGCAAATAATATCG |
| ChR forward | AAGGGAGCTGCAGTGGAGTA |
| ChR reverse | CCGAAAATCTGTGGGAAGTC |
| ChR mut forward | ACATGGTCCTGCTGGAGTTC |
| ChR mut reverse | GGCATTAAAGCAGCGTATCC |

aCSF at 22°C for the duration of the experiment. All solutions were bubbled with carbogen gas [95% $O_2$/ 5% $CO_2$].

## Organotypic brain slices

*Neonatal* mice (P6-8) were sacrificed by cervical dislocation followed by decapitation. The brains were extracted in ice-cold dissection medium (487.5 ml Earle's balanced salt solution [EBSS] and 12.5 ml of 25 mM HEPES). Three hundred μm thick transverse hippocampal slices were cut using a vibratome (Leica VT1200S) in ice-cold dissection medium. Organotypic slices were prepared using the Stoppini interface method as described in *De Simoni and Yu, 2006*; *Stephen et al., 2015*; *Stoppini et al., 1991*. Briefly, the slices were kept on sterile 0.45 μm Omnipore membrane inserts (Millipore, cat no. FHLC01300) in an incubator (37°C, 95% $O_2$/5% $CO_2$) for at least 6 days in culture media (47% MEM + GlutaMAX, 25% horse serum, 25% EBSS supplemented with 20 mM HEPES, 1.44% of 45% glucose, 1.06% penicillin/streptomycin with 16% nystatin, and 0.5% 1 M Tris solution) prior to imaging. The media were changed on the day of slicing and half of the media were replaced with fresh media every 3–4 days.

## Microscopy

### Fixed confocal imaging

Confocal images (1024 × 1024) were acquired on a Zeiss LSM700 upright confocal microscope using the 10× air, 20× water, and 63× oil objective and digitally captured using the default LSM acquisition software. For analysis, two to three zoomed regions of the hippocampus were imaged with the 2× zoom. For the quantification, these regions were averaged and represented as one value. Acquisition settings and laser power were kept constant within experiments. For neuronal reconstructions, ~150 μm thick confocal stacks were captured using the 20× objective, with z-steps of 1 μm.

### Live two-photon imaging

MitoDendra+ mitochondria in neuronal processes (excluding somatic mitochondria) in organotypic slices were live-imaged using the 20× and 60× water objectives on a Zeiss LSM700 upright two-photon microscope equipped with a MaiTai Ti:Saphire Laser (Spectra-Physics). The slices were transferred to a recording chamber, perfused with aCSF (2 mM $CaCl_2$, 2.5 mM KCl, 1 mM $MgCl_2$, 10 mM D-glucose, 126 mM NaCl, 24 mM $NaHCO_3$, 1 mM $NaH_2PO_4$) bubbled with carbogen gas and heated between 35 and 38°C at a constant perfusion (~2 ml/min). The excitation wavelength was set at 900 nm, and the rate of image acquisition was one frame/5 s for 500 s (100 frames/movie).

**Tissue processing and labeling**

## Brain harvesting

Adult animals (>P60) were sacrificed by cervical dislocation or $CO_2$ exposure. The brains were dropped-fixed in 4% paraformaldehyde (PFA) in sucrose solution overnight at 4°C. The brains were then cryo-protected in 30% sucrose/1× phosphate-buffered saline (PBS) (1.37 mM NaCl, 2.7 mM KCl, 10 mM $Na_2HPO_4$, 2 mM $KH_2PO_4$) solution overnight at 4°C before freezing at −80°C. Hemifloxed and conditional knock-out animals expressing the MitoDendra fluorophore were anaesthetized with isoflurane and transcardially perfused with ice-cold 4% PFA to maintain mitochondrial morphology. The frozen brains were embedded in tissue freezing compound (OCT), and 30 μm coronal brain slices were serially cryosected using a Cryostat (Bright Instruments). After live-imaging and electrophysiological recordings, brain slices were fixed in 4% PFA/sucrose solution overnight at 4°C. Slices were either kept in PBS at 4°C for short-term storage or at −20°C in cryoprotectant solution (30% glycerol, 30% ethylene glycol, 40% 1× PBS) for long-term storage.

## Immunohistochemistry

Free floating sections were washed with 1× PBS and permeabilized for 4–5 hr in block solution (1× PBS, 10% horse serum supplemented with 0.02% sodium azide, 3% [w/v] bovine serum albumin [BSA], 0.5% Triton X-100, and 0.2 M glycine). The slices were further blocked overnight with an added purified goat anti-mouse Fab-fragment (50 μg/ml, Jackson Immunoresearch) for reducing endogenous background. The sections were then incubated with primary antibody diluted in block solution overnight at 4°C. The following primary antibodies were used: Parvalbumin (mouse, 1:500, Millipore MAB1572), COX-IV (rabbit, 1:500, Abcam ab16056), Rhot1 (rabbit, 1:100, Atlas HPA010687), Bassoon (mouse, 1:500, Abcam ab82958), and Homer (rabbit, 1:500, Synaptic Systems 160 002). Slices were washed four to five times in 1× PBS over 2 hr and then incubated for 3–4 hr with secondary antibody in block solution (1:500-1:1000) at room temperature. The secondary antibodies used were the donkey anti-mouse Alexa Fluor 488 (Jackson ImmunoResearch 715-545-151), goat anti-rabbit Alexa Fluor 555 (Thermo Fisher Scientific A-21430), and donkey anti-rabbit Alexa Fluor 568 (Thermo Fisher Scientific A-10042). The slices were then washed four to five times in PBS for 2 hr and mounted onto glass slides using Mowiol mounting media. For the excitatory synapse staining, the free floating sections were subjected to an antigen retrieval protocol before permeabilization and antibody staining (*Jiao et al., 1999*). Briefly, the slices were incubated with preheated 10 mM sodium citrate buffer (pH 8.5) for 30 min at 80°C. The slices were cooled to room temperature in sodium citrate buffer and washed four to five times for 5 min with PBS.

## Biocytin labeling

Biocytin-filled slices were fixed in 4% PFA solution after intracellular recordings and kept overnight at 4°C. The slices were washed with 1× PBS three to four times and permeabilized with 0.3%-Triton 1× PBS for 4–5 hr. Streptavidin conjugated to Alexa Fluor 555 (Invitrogen S32355) in PBS-T 0.3% (1:500) was added, and the slices were kept overnight at 4°C. The slices were then washed four to five times in 1× PBS over 2 hr. The slices were placed on glass slides, and a coverslip was mounted on top using Dako Fluorescent mounting medium.

**Image analysis**

## Mitochondrial trafficking

The image sequences were subjected to alignment (stackreg) if necessary, background subtraction (rolling ball radius = 50 pixels) and filtering (smooth filter). Moving mitochondria were visually identified. Mitochondria were manually tracked between each frame using MTrackJ (*Meijering et al., 2012*) on Fiji, which provided track statistics. Mitochondrial movement was usually accompanied by brief periods of immobility so data were omitted from the velocity ($\mu m.s^{-1}$) calculations when a mitochondrion was immobile for a period longer than 10 s. Mitochondria were considered mobile if the distance covered was longer than 2 μm in 5 min.

### Fluorescent intensity

The fluorescence intensities of COX-IV and *Rhot1* were quantified on Fiji. The signal from the parvalbumin channel was thresholded using the default settings, the parvalbumin cell body was selected using the wand tool, and a mask was generated that was then superimposed on the channels of interest to selectively record the fluorescent signal within the masked region.

### Excitatory synapses

For analyzing the Bassoon/Homer synaptic pairs, the Synapse Counter plugin for ImageJ was used (*Dzyubenko et al., 2016*). 1024 × 1024 high-magnification images (zoom 2×) were auto-thresholded using the Otsu Thresholding method. The rolling ball radius (background subtraction) and maximum filter parameters were set to 7 and 1, respectively. Default colocalization settings were used that accept 33–100% overlap between pre- and post-synaptic markers.

### Neuronal reconstruction

The biocytin signal in acute brain slices was used to manually reconstruct parvalbumin interneurons in Neuromantic (*Myatt et al., 2012*) and generate. SWC files for further analysis.

### Sholl analysis

3D Sholl analysis was performed on the .SWC file using the Matlab script described in *Madry et al., 2018*. To perform the 3D-MitoSholl, the volume of the .SWC file was filled out and a stack-mask was generated in the Simple Neurite Tracer Plugin on Fiji (*Longair et al., 2011*). To isolate mitochondria selectively in parvalbumin interneurons, the background was subtracted (rolling ball radius = 50 pixel) and the median filter was applied prior to binarization. The stack of the mitochondrial distribution in individual cells was generated by adding the stack-mask to the mitochondrial channel using the 'AND' function of the Image Calculator in Fiji. 3D-MitoSholl analysis was performed using a custom Matlab script, which quantified the number of MitoDendra pixels within each sholl ring, radiating out from the soma at 1 μm intervals.

### Proximity analysis

High-magnification confocal stacks of the parvalbumin interneuron axon were acquired from 350 μm slices for calculating the minimum distance between mitochondria and branch points. The minimum distance between branch points and mitochondria was performed on 3D confocal stacks. The mitochondrial confocal stack was binarized using the default method. The (x, y, z) coordinates of the branch points were specified, and the minimum distance between the branch points coordinates and the closest mitochondrion was calculated by measuring the Euclidean distance between the x, y, z coordinates and the first pixel encountered.

## Electrophysiology

### Extracellular recordings

Local field potentials (LFPs) of CA3 network oscillations were recorded in an interface recording chamber as described in *Antonoudiou et al., 2020*. Briefly, an extracellular borosilicate glass electrode (tip resistance 1–5 MΩ) was filled with aCSF (126 mM NaCl, 3.5 mM KCl, 2 mM $MgSO_4.7H_2O$, 1.25 mM $NaH_2PO4$, 24 mM $NaHCO_3$, 2 mM $CaCl_2$, and 10 mM glucose; osmolality 300 ± 10 mOsmol/kg) and was placed in hippocampal area CA3. γ-oscillations (20–80 Hz) were induced by perfusion of carbachol (5 μM) in carbogen-bubbled aCSF. Data were acquired using 10 kHz sampling rate and amplified x10 by Axoclamp 2A (Molecular Devices). The signal was further amplified ×100 and low-pass filtered at 1 kHz (LPBF-48DG, NPI Electronic). The signal was digitized at 5 kHz by a data acquisition board (ITC-16, InstruTECH) and recorded on Igor Pro 6.3 software (Wavemetrics). For the fEPSP experiments, slices were superfused at room temperature with aCSF (125 mM NaCl, 25 mM $NaHCO_3$, 25 mM glucose, 2.5 mM KCl, 1.25 mM $NaH_2PO_4$, 2 mM $CaCl_2$, and 2 mM $MgCl_2$) in a recording chamber and visualized with an Olympus BX 51WI microscope (Olympus Europa Holding GmbH, Hamburg, Germany) connected to a KPM-3 Hitachi infrared video camera. Glass microelectrodes were filled with aCSF and placed on the stratum radiatum of CA1 pyramidal neurons. A bipolar stimulating electrode (FHC Inc, Bowdoin, ME) was positioned under low magnification (10×) on Schaffer collateral afferent fibers. All solutions were bubbled with carbogen gas (95% $O_2$/ 5% $CO_2$).

## Intracellular recordings

Intracellular recordings were performed in a submerged chamber (32–33℃) using borosilicate glass pipettes (5–12 MΩ). Data were acquired through the MultiClamp 700B amplifier (Molecular Devices) and digitized at 10 kHz (ITC-18, InstruTECH). Acquisition of electrophysiological signals was performed using Igor Pro 6.3 (Wavemetrics). The signals were low-pass filtered at 10 kHz and 3 kHz for current-clamp and voltage-clamp, respectively. For optogenetic experiments, filtered white LED (460 ± 30 nm, THOR labs) was delivered via epi-illumination through a 60× objective and was used to activate ChR2. For whole-cell current-clamp recordings, pipettes were filled with internal solution (110 mM K-gluconate, 40 mM HEPES, 2 mM ATP-Mg, 0.3 mM GTP-NaCl, 4 mM NaCl, 3–4 mg/ml biocytin; pH ~7.2; osmolality 270–290 mOsmol/kg). After break through, the bridge balance was adjusted to compensate for electrode access. Hyperpolarizing and depolarizing square current pulses were applied in order to quantify intrinsic properties of the recorded neuron. sEPSCs on parvalbumin interneurons were recorded in voltage clamp mode at −70 mV, using the current clamp internal solution. IPSCs on pyramidal cells were recorded in voltage clamp mode with pipettes filled with internal solution (137 mM CsCl, 5 mM NaCl, 10 mM HEPES, 0.1 mM EGTA, 2 mM ATP-Mg, 0.3 mM GTP-Na, 3–4 mg/ml biocytin). These experiments were done on two batches of animals. For evoked inhibitory post-synaptic currents (eIPSCs), the cells were held at −40 mV. This was to prevent spiking during light illumination in the absence of QX-314. In the second set of recordings, 5 mM QX-314 was added and the voltage was still held at −40 mV. The recordings between the first and second batches were comparable and therefore pooled together and quantified as one dataset. For optogenetic experiments, aCSF was supplemented with 3 mM kynurenic acid. After breakthrough, the cell and electrode capacitance were compensated in Multiclamp. Series resistance (RS) compensation was performed to 65%. For optogenetic experiments, a power plot using increasing light intensities was generated to decide the level of LED voltage to be used. The LED voltage that elicited 90% of the maximal response was used for stimulations. The LED output range used for power plot was between 0 and 1.53 mW.

## Electrophysiological analysis

In order to characterize and analyze the γ-oscillations, we calculated power spectra as the normalized magnitude square of the FFT (Igor Pro 6.3). The 50 Hz frequency was not included in the analysis to exclude the mains noise. The oscillation amplitude was quantified by measuring the peak of the power spectrum (peak power) and the area below the power spectrum plot (power area) within the γ-band range (20–80 Hz). The peak frequency of the oscillation was the frequency at which the peak of the power spectrum occurred in the γ-band range. For testing rhythmicity, autocorrelation was computed in Igor Pro 6.3. The peak power, power area, and peak frequency were calculated for a period of 300 s and compared between control, hemi-floxed, and KO mice. The power spectra were also calculated over a period of 300 s and a gaussian curve fit was performed using Igor Pro 6.3. This was done in order to assess the width of the power spectrum distribution. For spontaneous post-synaptic current (sPSC) detection, custom written procedures were used in Igor Pro 6.3. The traces were first low-pass filtered at 1 kHz. sPSCs were first detected using an initial threshold of 3 pA. The detected events that were smaller than 5× standard deviation of noise were excluded from further analysis. The traces were then visually inspected for correct peak identification. The traces were averaged, and the median inter event interval and peak amplitude were obtained for each cell. For the recovery experiments, the percentage of recovery was normalized to the amplitude of the first peak in the train, for every train and was presented as an average of the 10 trains. For the fEPSP experiments, stimulation was applied by the bipolar stimulating electrode (FHC Inc, Bowdoin, ME) every 10 s using constant current (0.2–1 µA, 80 µs square pulses). To set the intensity of stimulation, an input–output (I/O) relation was obtained for each slice when applying the control perfusion solution. The stimulus intensity was set such that the amplitude of the test fEPSP reached around 40% of maximum amplitude based on the I/O curve.

## Animal behavior testing

### Nesting assesment

The assessment husbandry behavior using Nestlets was performed as described in *Deacon, 2006*. Briefly, 1 hr before the dark cycle, mice were transferred in single cages containing wood-chip

bedding but no other environmental enrichment. One Nestlet (5 × 5 cm) was placed in each cage. The following morning, the shredded and unshredded Nestlets were collected and weighted.

### Open field

The open-field apparatus consisted of a 45 × 45 × 45 cm box open at the top. Mice were placed in the center of the arena and were recorded for 10 min via a ceiling-fixed video camera. The movies were analyzed using ToxTrac, an automated open-source executable software (*Rodriguez et al., 2018*). Detection settings were set to 10 for threshold, 100 for minimum object size, and 1000 for maximum object size, and the default tracking settings were used.

### Rotarod

The assessment of motor coordination and learning was performed similar to *Deacon, 2013*. Briefly, mice were placed on a revolving rotarod treadmill (Med Associates) with a starting speed of 4 revolutions per minute (RPM) and an acceleration rate of about 7 RPM/min. Each mouse was subjected to a 5 min trial, repeated two more times after a 10 min gap. To calculate the RPM, we recorded the speed at which the mouse fell from the revolving rod using this formula: RPM = [(End Speed − Start Speed)/300] × (Seconds Run) + Start Speed.

### T-Maze

The assessment of spontaneous alternation on the T-maze was performed according to *Deacon and Rawlins, 2006*. The T-maze apparatus consisted of three arms with 27 × 7 × 10 cm dimensions. A set of three sliding guillotine doors was used to separate the entrance of each arm. At the beginning of the experiment, all of the doors were raised, except for the one located in front of the starting point. Each mouse was allowed eight consecutive trials. Each trial consisted of individually placing each animal at the entrance of the main arm and allowing it to freely run and chose an arm. After the mouse entered an arm, the guillotine door was pushed down and the animal was confined in the chosen arm for 30 s. The mouse was then placed in the starting point for 30 s before the beginning of the next trial. Each trial lasted less than 90 s. When a mouse did not choose an arm within 90 s, it was considered as a NO-GO and the trial was restarted. Animals were excluded from the quantification when there were more than 3 NO-GOs.

### Elevated plus maze

The elevated plus maze was placed 40 cm above the ground and consisted of four 30 × 5 cm arms, two of which were surrounded by additional 15 cm high walls. Mice were placed on the boundary between the open arm and the center, facing the center. Mice were recorded for 5 min by a ceiling-fixed video camera, and the movies were analyzed using ToxTrac (*Rodriguez et al., 2018*).

## Statistical analysis and blinding

Statistical analysis was performed in Prism 6, Excel, and Igor Pro. Datasets were tested for fitting in a Gaussian distribution using the D'Agostino–Pearson omnibus and the Shapiro–Wilk normality test. When the distribution was normal, unpaired t-test was performed. The F-test was used to compare variances. When the variance was significantly different, the unpaired t-test with Welch's correction was performed. For cumulative distributions, the Kolmogorov–Smirnov (KS) test and multiple t-test per row were performed. For non-normally distributed data, the Mann–Whitney (MW) test was applied. For comparing multiple groups, ordinary one-way ANOVA and Tukey's multiple comparisons test with a single pooled variance were used. Statistical outliers were identified using the ROUT method and removed (*Motulsky and Brown, 2006*). Significance of $p < 0.05$ is represented as *, $p < 0.01$ as **, $p < 0.001$ as ***, and $p < 0.0001$ as ****. The errors in all bar charts represent the standard error of the mean (sem). The box-and-whisker plots (boxplots) represent the min to max with the middle line plotted at the median. Brain tissue processing, staining, acquisition, and analysis were performed blinded. Live imaging acquisition and analysis were performed blinded for littermate hemi-floxed and conditional knock-out animals. Electrophysiological recording analysis was performed blinded.

## Acknowledgements

This work was supported by a PhD studentship (1405150) from the Medical Research Council (MRC) to GK (MRC LMCB 4-year PhD Programme) and grants from the MRC (MR/M024083/1) to PCS, the European Research Council grant 282430 (Fuelling Synapses), and the Lister Institute of Preventive Medicine to JTK.

## Additional information

### Funding

| Funder | Grant reference number | Author |
| --- | --- | --- |
| Medical Research Council | 1405150 | Georgina Kontou |
| Medical Research Council | MR/M024083/1 | Patricia C Salinas |
| European Research Council | 282430 | Josef T Kittler |
| Lister Institute of Preventive Medicine | | Josef T Kittler |

The funders had no role in study design, data collection and interpretation, or the decision to submit the work for publication.

### Author contributions

Georgina Kontou, Conceptualization, Data curation, Formal analysis, Validation, Investigation, Methodology, Writing - original draft, Writing - review and editing; Pantelis Antonoudiou, Marina Podpolny, Blanka R Szulc, Investigation, Methodology, Writing - review and editing; I Lorena Arancibia-Carcamo, Methodology, Writing - review and editing; Nathalie F Higgs, Guillermo Lopez-Domenech, Supervision, Writing - review and editing; Patricia C Salinas, Resources, Funding acquisition, Writing - review and editing; Edward O Mann, Conceptualization, Resources, Supervision, Writing - review and editing; Josef T Kittler, Conceptualization, Funding acquisition, Project administration, Writing - review and editing

### Author ORCIDs

Georgina Kontou https://orcid.org/0000-0002-0551-1577
I Lorena Arancibia-Carcamo http://orcid.org/0000-0002-0624-3850
Guillermo Lopez-Domenech http://orcid.org/0000-0002-3114-2082
Patricia C Salinas http://orcid.org/0000-0002-5748-083X
Edward O Mann https://orcid.org/0000-0002-2468-7148
Josef T Kittler https://orcid.org/0000-0002-3437-9456

### Ethics

Animal experimentation: All experimental procedures were carried out in accordance with institutional animal welfare guidelines and licensed by the UK Home Office in accordance with the Animals (Scientific Procedures) Act 1986.

### Decision letter and Author response

Decision letter https://doi.org/10.7554/eLife.65215.sa1
Author response https://doi.org/10.7554/eLife.65215.sa2

## Additional files

### Supplementary files

- Source code 1. MitoSholl Matlab Script.
- Transparent reporting form

## Data availability

All data generated or analysed are included in the manuscript, supporting files and source data. The neuronal reconstruction data have been deposited to the NeuroMorpho database (http://neuromorpho.org/NeuroMorpho_Linkout.jsp?PMID=34190042).

The following dataset was generated:

| Author(s) | Year | Dataset title | Dataset URL | Database and Identifier |
|---|---|---|---|---|
| Kontou G, Antonoudiou P, Podpolny M, Szulc BR, Arancibia-Carcamo IL, Higgs NF, Lopez-Domenech G, Salinas PC, Mann EO, Kittler JT | 2021 | Miro1-dependent Mitochondrial Dynamics in Parvalbumin Interneurons | http://neuromorpho.org/NeuroMorpho_Linkout.jsp?PMID=34190042 | NeuroMorpho, 34190042 |

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
