## [Decision Letter]

**Acceptance summary:**

The manuscript dissects the role of a key cellular organelle, mitochondria, in cellular structure and functioning of fast-spiking parvalbumin-positive (PV+) interneurons. The study demonstrates that the conditional removal in PV+ cells of Miro1, a protein involved in mitochondrial mobility, results in accumulation of mitochondria near the nucleus, reduced numbers in synaptic terminals, and impaired axonal mitochondrial trafficking, localization and axonal branching. Intriguingly, such changes in PV+ cell mitochondrial density and/or axonal branching were associated with faster γ-oscillations and reduced anxiety.

**Decision letter after peer review:**

Thank you for submitting your article "Miro1-dependent Mitochondrial Dynamics in Parvalbumin Interneurons" for consideration by *eLife*. Your article has been reviewed by 2 peer reviewers, and the evaluation has been overseen by a Reviewing Editor and John Huguenard as the Senior Editor. The reviewers have opted to remain anonymous.

Summary:

By specifically deleting Miro1, a protein involved in mitochondrial mobility, in parvalbumin (PV)-expressing neurons, the authors explore the impact of mitochondrial mobility and localization on the cellular properties of PV cells, their synaptic inputs and outputs, the impact on network properties in the hippocampus and mouse behavior. The authors utilize cre-lox recombination to delete Miro1, and in parallel to visualize mitochondria or express channel-rhodopsin, specifically in PV neurons. The authors find that Miro1 deletion reduces but does not arrest completely mitochondrial mobility, alters the distribution of mitochondria in the axon, specifically changes the arborization pattern of the axon of PV neurons (but not of dendrites), affects only slightly the resting properties of PV neurons and does not affect their evoked synaptic properties. In contrast, they observed that the excitatory input onto PV cells was enhanced, and that carbachol-induced network oscillation (in the hippocampus) shifted to a higher frequency and were more variable. Lastly, they report that anxiety-related behavior in these mice was reduced.

Overall, this is a well-executed study supported by high quality ex vivo data. The study potentially advances our understanding as how axonal mitochondrial trafficking and positioning could shape hippocampal network activity. In particular, the descriptive portion of the phenotypes is well done and solid. However, the mechanistic insights into altered excitatory input on PV cells and network activity are insufficient and should be extended.

Essential Revisions:

1. We expect the authors to provide an explanation why the excitatory inputs onto the PV cells are enhanced (frequency of minis and their amplitude)? Are there changes in the numbers of excitatory inputs? Is this alteration specific to PV cells, or is the excitatory tone in these mice enhanced altogether?

2. The authors show that the redistribution of mitochondria does not affect the properties of synaptic contacts of PV cells on peri-somatic neurons. They suggest this is because the functions that mitochondria perform, such as ATP production and roles in calcium handling, may not be perturbed. They suggest that diffusion of ATP may be sufficient to support synaptic function away from mitochondrial ATP sources. Both of these arguments are not well substantiated. First, there is evidence that Miro1 participates in mitochondrial quality control (10.1016/j.cell.2011.10.018), suggesting that it is of interest to determine if the function of mitochondria in PV neurons in this study is preserved. Second, while the authors cite literature that shows that mitochondrial presence in synapses is not critical for their function (Pathak et al., 2015), there are several papers indicating that this is not the case (for example, 10.1016/j.celrep.2013.06.040: cited by the authors, but not for this purpose). The papers by Ashrafi et al. (2017, 2019) should also be discussed in this context. Finally, except for stating that mitochondria have a role in calcium handling, there is no discussion of this topic. Could it be that PV, as a calcium buffer, can compensate for the redistribution of mitochondria, at least partially?

3. The authors do not directly address the question of the effect of the deletion of Miro1 in PV cells on their viability. Indeed, the authors suggest that because the PV promotor becomes active after P10, viability of PV neurons should not be compromised, but this is not substantiated. They authors should directly assess this question. They could count PV neurons in the relevant brain areas at the time when their experiments are conducted. This should be an easy and important point to establish.

4. Figure 1: these data further validate the well-known function of Miro1 in driving axonal mitochondrial transport in these specific PV+ interneurons. However, it is difficult to see altered mitochondrial motility by comparing individual mitochondrion motility marked by arrows in the images of the three genotypes. Kymographs generated from selected axonal branches would be much more helpful.

5. Figure 2: while biocytin filling helps with displaying neuronal morphologic details in live brain slices, the authors should confirm whether the majority of biocytin-filled puncta represent presynaptic boutons, but not axonal varicosities with increased volume or branch points (Figure 3I). In addition to compare mitochondria-containing axonal boutons, I would suggest the authors characterize boutons by quantifying their relative mitochondrial mean intensity within biocytin-filled puncta. It is my assumption that much less mitochondria are clustered in presynaptic boutons in Miro1 KO PV+ interneurons if they compare each individual bouton. This quantitative data would likely be more robust in supporting their conclusion. Furthermore, the data showing the minimum distance between boutons and mitochondria does not make much sense given the fact that mitochondrial density is significantly declined in mutant axons. One would argue whether Miro1 plays a secondary role in recruiting or retaining presynaptic mitochondria in vivo, although the authors discuss this possibility but do not further explore it in the study.

6. Figure 3: while reconstruction of the PV+ interneurons convincingly displays increased axonal branching in the hippocampal parvalbumin with Miro1 deletion, it is rather challenging to determine axon branch points in brain slices only by using biocytin filling. How could the authors confidently localize axonal branch points versus presynaptic boutons under limited resolution? Thus, the data characterizing minimum distance between a branch point and a mitochondrion in the confocal stack is not quite convincing.

7. Figure 4: selective photoactivation of PV+ interneurons is elegant. Given the fact that 51% of presynaptic boutons retain mitochondria in Miro1 KO PV+ interneurons (Figure 2G), it is still higher than normal presynaptic boutons in hippocampal CA3 regions (Smith et al., *eLife* 2016). It is likely that such mild reduction in presynaptic mitochondria would not significantly affect sIPSCs and short-term recovery of the inhibitory responses. The authors have postulated that PV+ interneurons require substantial amounts of energy to sustain the high firing rate during neuronal transmission. What would be the effect of increasing stimulation time frame from 2 sec to 10 or 30 sec? Strong and prolonged synaptic stimulation would more likely deplete presynaptic energy when presynaptic boutons don't retain a mitochondrion, thus energy-dependent phenotypes might be more robustly affected.

---

## [Author Response]

Essential Revisions:1. We expect the authors to provide an explanation why the excitatory inputs onto the PV cells are enhanced (frequency of minis and their amplitude)? Are there changes in the numbers of excitatory inputs? Is this alteration specific to PV cells, or is the excitatory tone in these mice enhanced altogether?

We would like to thank the reviewers for their suggestion. To address whether the increase in excitation received by parvalbumin (PV+) interneurons was specific or a general enhancement in excitatory tone, we performed immunohistochemical (IHC) experiments on fixed brain tissue of control (WT: Miro1^(+/+)^) and conditional knock-out (KO: Miro1^(Δ/Δ)^) mice. The aim of this experiment was to assess whether the increased excitation received by PV+ interneurons was accompanied by structural changes in excitatory synapses in the hippocampus. To visualize excitatory synapses, we immuno-stained with antibodies against the pre-synaptic marker Bassoon and the post-synaptic marker Homer (Figure 5—figure supplement 1A). We observed no significant differences in the total immuno-stained area that each synaptic marker occupied, or in the Bassoon-Homer overlap (colocalization). Thus, the loss of Miro1 from PV+ interneurons does not seem to affect the overall excitatory synapse levels in the hippocampal strata. Next, we performed extracellular recordings of field excitatory postsynaptic potentials (fEPSP) at increasing electrical stimulations (Figure 5—figure supplement 1B). The input-output (I-O) relationship indicated that basal synaptic transmission was higher in the absence of Miro1 compared to the control, suggesting an increase in the overall excitatory tone in these brain slices. Even though the total number of excitatory synapses did not change, excitatory transmission seems to be potentiated in the Miro1 KO (0.19 ± 0.033 a.u.) compared to the control (0.10 ± 0.005 a.u.), as demonstrated by the increase in the fEPSP slope at high stimulations (Figure 5—figure supplement 1B, *p* = 0.0018, Tukey’s multiple comparisons test, ordinary one-way ANOVA F (3, 139) = 17.58, P < 0.0001). These data suggest that the hippocampal network is more excited in the absence of Miro1 from PV+ interneurons.

2. The authors show that the redistribution of mitochondria does not affect the properties of synaptic contacts of PV cells on peri-somatic neurons. They suggest this is because the functions that mitochondria perform, such as ATP production and roles in calcium handling, may not be perturbed. They suggest that diffusion of ATP may be sufficient to support synaptic function away from mitochondrial ATP sources. Both of these arguments are not well substantiated. First, there is evidence that Miro1 participates in mitochondrial quality control (10.1016/j.cell.2011.10.018), suggesting that it is of interest to determine if the function of mitochondria in PV neurons in this study is preserved. Second, while the authors cite literature that shows that mitochondrial presence in synapses is not critical for their function (Pathak et al., 2015), there are several papers indicating that this is not the case (for example, 10.1016/j.celrep.2013.06.040: cited by the authors, but not for this purpose). The papers by Ashrafi et al. (2017, 2019) should also be discussed in this context. Finally, except for stating that mitochondria have a role in calcium handling, there is no discussion of this topic. Could it be that PV, as a calcium buffer, can compensate for the redistribution of mitochondria, at least partially?

Thank you for your comment. To investigate whether the function of mitochondria in PV+ neurons is preserved in the absence of Miro1, we bulk-loaded organotypic brain slices with Tetramethyl-rhodamine, methyl ester (TMRM) and live-imaged them under a confocal microscope (Author response image 1). TMRM is a cell permeable fluorescent dye that is sequestered by active mitochondria. The uptake of TMRM by mitochondria is proportional to the membrane potential (ΔΨ) (Scaduto and Grotyohann, 1999). We observed no difference in the mean TMRM intensity in control and cKO cells (WT 1024 ± 108.8 a.u., Miro1 KO 1036 ± 182.8 a.u., Mann Whitney U test NS *p* = 0.113), suggesting that the respiratory properties of mitochondria do not change when Miro1 is knocked-out. Consistent with this observation, similar experiments in mouse embryonic fibroblast (MEF) lines and dissociated primary motor neuron cultures also showed that TMRM intensity was not affected by the loss of Miro1 (Shaw et al., 2014; López-Doménech et al., 2016). Initially, we expected that the alteration in mitochondrial positioning would be sufficient to see changes in Ca^2+^ buffering, directly affecting neurotransmitter release, represented as increased facilitation. Surprisingly, we observed no difference in the paired-pulse ratio (PPR) of the first two pulses of the train during photostimulation (Author response image 1; WT 0.99 ± 0.073, Miro1 KO 0.95 ± 0.041, *p* = 0.6512, unpaired t-test with Welch’s correction). As the reviewer suggested, one possibility could be that parvalbumin behaves as a Ca^2+^ buffer in the absence of a stable mitochondrial buffering pool. The Ca^2+^ may be sequestered by parvalbumin itself and no longer participate in synaptic release. The fluorescent levels of parvalbumin were slightly increased in Miro1 KO compared to WT and may compensate for the altered mitochondrial positioning (Author response image 1; WT 79 ± 4.9 a.u., Miro1 KO 99 ± 9.2 a.u., *p* = 0.02 Mann Whitney U test). We have now updated the discussion to include a comment on Ca^2+^ buffering by parvalbumin and expanded the discussion to include the papers suggested by the reviewers.

**Author response image 1. sa2fig1:** Parvalbumin levels seem to be increased in the Miro1 KO while mitochondrial TMRM fluorescence and synaptic facilitation are unaffected by the absence of Miro1. (**A**) Loss of Miro1 does not affect mitochondrial TMRM uptake in parvalbumin interneurons. Representative confocal images from control and cKO organotypic brain slices that were bulk loaded with TMRM. The MitoDendra signal was used as a mask to isolate the TMRM signal emerging from mitochondria in parvalbumin interneurons. Scale Bar = 10 μm. Boxplot shows the quantification for the mean TMRM fluorescence in the cell (n_WT_ = 28 neurons from 6 slices from 3 animals, n_KO_ = 21 neurons from 4 slices from 2 animals). (**B**) Loss of Miro1 does not alter short-term facilitation. Example responses from control and Miro1 KO cells in the hippocampus. The bar chart shows the quantification for the mean paired-pulse ratio (PPR) of the second response divided by the first response to light and the error bars represent the standard error of the mean (n_WT_ = 19, n_KO_ = 22 recordings from 4 animals). (**C**) Loss of Miro1 increases parvalbumin levels in the hippocampus. Boxplot and cumulative frequency distribution of the parvalbumin fluorescent intensity signal (n_WT_ = 79 neurons from 10 slices from 3 animals, n_KO_ = 73 neurons from 8 slices from 3 animals).

3. The authors do not directly address the question of the effect of the deletion of Miro1 in PV cells on their viability. Indeed, the authors suggest that because the PV promotor becomes active after P10, viability of PV neurons should not be compromised, but this is not substantiated. They authors should directly assess this question. They could count PV neurons in the relevant brain areas at the time when their experiments are conducted. This should be an easy and important point to establish.

We would like to thank the reviewers for the suggestion. We have counted the number of PV+ neurons in acute hippocampal slices from the WT, ΗΕΤ, and Miro1 KO mice (Figure 1 —figure supplement 1 ). As expected, we do not see a significant difference in the number of MitoDendra-expressing PV+ interneurons, supporting the argument that the PV cell viability is not compromised in our transgenic mouse models (WT 85.25 ± 5.487, ΗΕΤ 75.71 ± 8.083, Μiro1 KO 70.56 ± 8.257 cells).

4. Figure 1: these data further validate the well-known function of Miro1 in driving axonal mitochondrial transport in these specific PV+ interneurons. However, it is difficult to see altered mitochondrial motility by comparing individual mitochondrion motility marked by arrows in the images of the three genotypes. Kymographs generated from selected axonal branches would be much more helpful.

We would like to thank the reviewers for the suggestion. We have now updated the figure to include kymographs and replaced colored arrows with the actual motion track of individual mobile mitochondria (Figure 1).

5. Figure 2: while biocytin filling helps with displaying neuronal morphologic details in live brain slices, the authors should confirm whether the majority of biocytin-filled puncta represent presynaptic boutons, but not axonal varicosities with increased volume or branch points (Figure 3I).

To confirm that the PV+ interneuron biocytin-filled varicosities along the axon were indeed presynaptic terminals, we performed an IHC experiment using antibodies against the inhibitory presynaptic marker vGAT (Author response image 2). The biocytin-filled puncta co-localized with the vGAT clusters suggesting that they are inhibitory presynaptic boutons along the axon. This observation is also supported by existing work in the literature (Gu et al., 2017).

**Author response image 2. sa2fig2:** Biocytin-filled boutons on the PV+ interneuron axon represent inhibitory pre-synaptic terminals. Example max-projected confocal images of biocytin-filled axon. Scale Bar = 3 μm. The graphs represent the fluorescence signal of vGAT (red) from a line-scan (white line) through a biocytin-filled bouton (green).

In addition to compare mitochondria-containing axonal boutons, I would suggest the authors characterize boutons by quantifying their relative mitochondrial mean intensity within biocytin-filled puncta. It is my assumption that much less mitochondria are clustered in presynaptic boutons in Miro1 KO PV+ interneurons if they compare each individual bouton. This quantitative data would likely be more robust in supporting their conclusion. Furthermore, the data showing the minimum distance between boutons and mitochondria does not make much sense given the fact that mitochondrial density is significantly declined in mutant axons. One would argue whether Miro1 plays a secondary role in recruiting or retaining presynaptic mitochondria in vivo, although the authors discuss this possibility but do not further explore it in the study.

We would like to thank the reviewers for the suggestion. We have now reanalyzed the data as suggested by the reviewer and quantified the mean intensity of MitoDendra within the biocytin-filled puncta. We report a reduction in the MitoDendra fluorescence within the biocytin-filled puncta in the Miro1 KO cells (WT 15.2 ± 0.64 a.u., Miro1 KO 9.2 ± 0.60 a.u. Author response image 3, *p* < 0.0001, Mann-Whitney U test). For quantification purposes, a biocytin filled cluster was defined as empty if the MitoDendra intensity was less than the background signal (<8 a.u., based on visual assessment). 70% of boutons in WT and 46% of the boutons in Miro1 KO contain MitoDendra+ signal. This observation is consistent with our existing data presented in Figure 2G, where 74 ± 4.1% of presynaptic terminals contained MitoDendra+ mitochondria in control cells and 51 ± 4.0% in the cells where Miro1 was knocked-out (*p* = 0.003). As suggested by the reviewers, we have now omitted the minimum distance data from the paper (Figure 2J) and included the new quantification in the main manuscript.

**Author response image 3. sa2fig3:** MitoDendra fluorescence within the biocytin-filled puncta is reduced when Miro1 is knocked-out from PV+ interneurons. Boxplot and cumulative distribution for the quantification of the mean MitoDendra fluorescent intensity within biocytin-filled puncta (n_Miro(+/+)_ = 246, n_Miro(Δ/Δ)_ = 173 boutons).

6. Figure 3: while reconstruction of the PV+ interneurons convincingly displays increased axonal branching in the hippocampal parvalbumin with Miro1 deletion, it is rather challenging to determine axon branch points in brain slices only by using biocytin filling. How could the authors confidently localize axonal branch points versus presynaptic boutons under limited resolution? Thus, the data characterizing minimum distance between a branch point and a mitochondrion in the confocal stack is not quite convincing.

The diffusion of biocytin is sufficient to distinguish axonal segments in PV+ interneurons and adequate in identifying branch points. To identify branch points, small stacks were isolated and bifurcation/branch/splitting points were identified by adjusting the brightness and contrast of the images. The images show examples of how branch points were defined, as opposed to segments where two axons were crossing, based on the biocytin signal (Author response image 4).

**Author response image 4. sa2fig4:** Identification of axonal branch points in biocytin-filled PV+ interneurons. Adjustment of brightness and contrast allows to distinguish between bifurcation points and points where the axons cross over each other..

7. Figure 4: selective photoactivation of PV+ interneurons is elegant. Given the fact that 51% of presynaptic boutons retain mitochondria in Miro1 KO PV+ interneurons (Figure 2G), it is still higher than normal presynaptic boutons in hippocampal CA3 regions (Smith et al., eLife 2016). It is likely that such mild reduction in presynaptic mitochondria would not significantly affect sIPSCs and short-term recovery of the inhibitory responses. The authors have postulated that PV+ interneurons require substantial amounts of energy to sustain the high firing rate during neuronal transmission. What would be the effect of increasing stimulation time frame from 2 sec to 10 or 30 sec? Strong and prolonged synaptic stimulation would more likely deplete presynaptic energy when presynaptic boutons don't retain a mitochondrion, thus energy-dependent phenotypes might be more robustly affected.

Increasing the stimulation time window to 5 seconds, has no effect on the ability of PV+ interneurons to sustaining long lasting inhibition after prolonged photo-stimulation (Author response image 5). The inhibitory responses reach plateau and effectively deplete in both control and cKO cells at similar rates. This data further supports that the cKO cells can sustain long lasting inhibition similarly to the control cells.

**Author response image 5. sa2fig5:** Cells in WT and Miro1 KO mice can sustain inhibition after long-lasting photostimulation. Example traces from the inhibitory responses pyramidal cells received in WT and Miro1 KO slices during light train stimulation (40 Hz for 5 s; 1 ms pulse width). The quantification shows the mean amplitude of each peak during the light train stimulation (n_WT_ = 4 recordings and n_KO_ = 7 recordings).